# FUNCTION-SPACE VARIATIONAL INFERENCE FOR DEEP BAYESIAN CLASSIFICATION

## ABSTRACT

Bayesian deep learning approaches assume model parameters to be latent random variables and infer posterior predictive distributions to quantify uncertainty, increase safety and trust, and prevent overconfident and unpredictable behavior. However, weight-space priors are model-specific, can be difficult to interpret and are hard to specify. Instead of weight-space priors, we leverage function-space variational inference to apply a Dirichlet predictive prior in function space, resulting in an implicit variational Dirichlet posterior which facilitates easier specification of predictive epistemic uncertainty. This is achieved through the perspective of stochastic neural network classifiers as variational implicit processes, which can be trained using function-space variational inference by devising a novel Dirichlet KL estimator. Experiments on small- and large-scale image classification tasks demonstrate that our function-space inference scales to large-scale tasks and models, improves adversarial robustness and boosts uncertainty quantification across models, without influencing architecture or model size.

## 1 INTRODUCTION

Deep learning (Goodfellow et al., 2016) has enabled powerful classification models capable of working with complex data modalities and scaling to large data sets (Krizhevsky et al., 2012), outperforming non-parametric models such as Gaussian processes (Rasmussen & Williams, 2005) in these domains. The aim of *Bayesian* neural networks (BNN) is to provide these complex models with function priors for regularization, generalization and uncertainty quantification (UQ) useful in prediction tasks (Gal, 2016; Wilson & Izmailov, 2020; Fortuin, 2021; Abdar et al., 2021). Predictive uncertainty is crucial for the deployment of machine learning systems in real-world settings, as it provides a degree of safety (McAllister et al., 2017), trust (Lim et al., 2019), sample efficiency (Deisenroth & Rasmussen, 2011; Gal et al., 2017) and human-in-the-loop cooperation (Filos et al., 2019). Unfortunately, the highly redundant and nonlinear parameterization of modern deep learning systems has been shown to be challenging for Bayesian methods, sometimes resulting in inadequate uncertainty quantification (Foong et al., 2020; 2019; Osband et al., 2018; Ovadia et al., 2019; Wenzel et al., 2020; Yao et al., 2019) even on toy tasks (e.g. Figure 3).

In this work, we consider function-space variational inference[1] (Sun et al., 2019) (fVI) for classification. This inference method reasons about priors in function space, specifying the prior with respect to their predictive distribution rather than their weight distribution. By approximately representing the posterior in function space using a variational implicit process (Ma et al., 2019), our approach can be used with several popular BNN representations such as deep ensembles (Lakshminarayanan et al., 2017) and Monte Carlo dropout (Gal & Ghahramani, 2016). By positing that the implicit predictive of a stochastic network classifier is Dirichlet distributed, rather than solely categorical, we gain a richer predictive uncertainty representation while retaining the same mean predictions as categorical classifiers. Furthermore, the approach provides a unifying perspective of prior work which use the Dirichlet distribution and function-space regularization (Malinin & Gales, 2018; Malinin et al., 2020; Joo et al., 2020; Sensoy et al., 2018; 2021), through a principled objective that does not require a specific model or data transformations. We demonstrate that this method improves uncertainty quantification and adversarial robustness across a range of popular models and datasets for both small- and large-scale inference.

---

[1] While prior work uses the term *functional* VI, we use function-space VI for clarity.

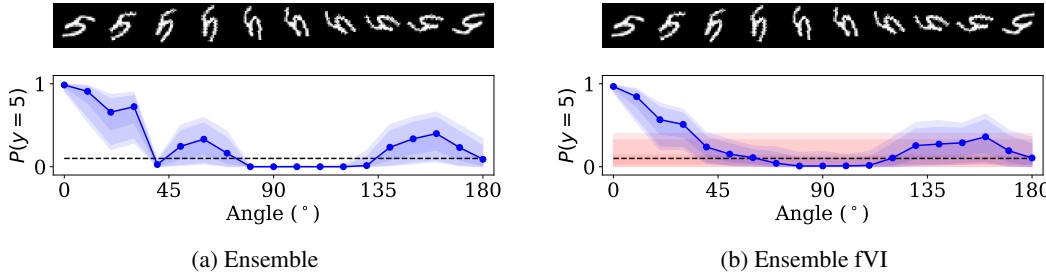

(a) Ensemble          (b) Ensemble fVI

Figure 1: Bayesian classification (blue) on the rotated MNIST task. Although the ensemble has reasonable uncertainty quantification, it can sporadically fluctuates over rotations and exhibits regions of overconfidence. Using function-space VI, the uniform prior (red) regularizes the prediction outside of the data distribution to be more consistent and reasonable in its uncertainty. Here, the index set is the training data and uncertainty is expressed through the 1st, 3rd and 25th percentile.

## 1.1 CONTRIBUTIONS

We incorporate Dirichlet predictive priors into deep Bayesian networks using function-space inference. Concretely, our contributions are:

1. We demonstrate an extension of variational implicit processes into the classification setting.
2. We propose a function-space variational inference method for classification, using an effective KL divergence estimator from softmax samples via Newton-based Dirichlet estimation.

## 2 RELATED WORK

We build on prior work that considers classification in the Bayesian setting and function-space inference of Bayesian neural networks.

## 2.1 BAYESIAN CLASSIFICATION

Compared to regression, classification is non-trivial for Bayesian methods due to the nonlinear link function required to predict the class labels. As a result, closed-form Bayesian models, such as Gaussian processes (GP), require approximate inference methods such as the Laplace approximation (Rasmussen & Williams, 2005), variational inference (Gibbs & Mackay, 2000; Hensman et al., 2015; Salimbeni et al., 2018; Izmailov et al., 2018), and expectation propagation (Hernandez-Lobato & Hernandez-Lobato, 2016). The Pólya-Gamma data augmentation trick (Polson et al., 2013) has enabled scalable closed-form variational training of sparse Gaussian process classifiers (Wenzel et al., 2019). Gaussian processes have also been used with a Dirichlet predictive using a log-normal approximation (Milios et al., 2018). Classification with Bayesian neural networks is possible through a wide range of approximate inference methods, including Markov chain Monte Carlo (Neal, 1995; Zhang et al., 2020), (mean-field) variational inference (MFVI) (Graves, 2011; Blundell et al., 2015; Zhang et al., 2018), Laplace approximations (MacKay, 1992; Denker & LeCun, 1991; Ritter et al., 2018; Khan et al., 2019; Immer et al., 2021; Daxberger et al., 2021), ensembles (Lakshminarayanan et al., 2017; Osband et al., 2018; Barber & Bishop, 1998; Pearce et al., 2020), expectation propagation (Hernández-Lobato & Adams, 2015) and Monte Carlo dropout (Gal & Ghahramani, 2016; Kingma et al., 2015). Radial BNNs (Farquhar et al., 2020) are motivated as a practical alternative to MFVI BNNs that uses Gaussian weight priors and posteriors. By sampling weights in a radial fashion, they avoid the pathologies encountered when sampling high-dimensional Gaussian distributions. Rank 1 BNNs (Dusenberry et al., 2020) combine ensembles and weight priors. Using the shared BatchEnsemble structure (Wen et al., 2020) and rank 1 covariance parameterizations, Rank 1 BNNs have a scalable memory requirement. Alternatively, the Laplace bridge (Hobbhahn et al., 2021) approximately maps a Dirichlet predictive density backwards through the softmax into a latent Gaussian predictive. A Gaussian-predictive BNN can then be trained using this latent approximation.

Alternative methods avoid propagating uncertainties by predicting Dirichlet concentrations directly with neural networks. Prior networks (Malinin & Gales, 2018; 2019) require categorical labels to be converted to Dirichlets, and mimic fVI as the objective consists of two KL divergences, for in- and outside the data distribution respectively. Another training method used prior networks to distill a

trained ensemble into a single model (Malinin et al., 2020). Similarly, belief matching (Joo et al., 2020) converts the labels to Dirichlets using Bayes rule. This method also echoes fVI, as the training objective is an fELBO where the index set is the training data. Another method converts the training labels to categorical probabilities and uses a Bayes risk objective with KL regularization against a function-space prior (Sensoy et al., 2018; 2021). Compared to these methods, we use fVI as a principled approach to inference that allows us to use standard BNN parameterizations with the categorical likelihood, avoiding the need to design networks and data representations that facilitate this training approach. For a longer discussion on this prior work, see Appendix A.

## 2.2 FUNCTION-SPACE VARIATIONAL INFERENCE

Function-space inference is best understood through Gaussian processes, which are closed-form distributions over functions. Using this view, sparse GPs may be viewed as approximate inference over functions (de G. Matthews et al., 2016), minimizing the fKL from its exact posterior via inducing points. The function-space variational BNN (fBNN) by Sun et al. (2019) uses the fKL to use explicit or implicit stochastic processes as function-space priors. They use a GP trained on the data as a prior, which can be viewed as a form of empirical Bayes. Similar approaches take a mirror descent view for batch training (Shi et al., 2019). However, function-space variational inference can yield ill-defined objectives in many cases Burt et al. (2021). Neural linear models have also been used with fVI, as they have an attractive closed-form Gaussian predictive (Watson et al., 2021a;b). Concurrent work by Rudner et al. (2021) has also adopted fVI for classification, by linearizing the network about a Gaussian weight distribution to estimate the fKL. This model works with a Gaussian (latent) predictive prior and posterior which loses the intuitive aspect of function-space priors. Moreover, the linearization requires computation of the weight Jacobian during training, for which the memory requirement scales with the number of model parameters and outputs. Variational implicit processes (Ma et al., 2019) take a stochastic process perspective of stochastic parametric regressors and, for the regression setting with Gaussian likelihoods, proposes a wake-sleep procedure for inference. Wang et al. (2019) propose particle optimization methods, using finite representations of functions to learn a particle representation of the function-space posterior through the gradient flow of the log posterior. Function-space inference is also an attractive approach to continual learning, e.g. Pan et al. (2020).

## 3 DIRICHLET PREDICTIVE PRIORS FOR BAYESIAN CLASSIFICATION

In this section, we discuss how function-space variational inference can be applied to classification using variational implicit processes (VIP) (Ma et al., 2019).

### 3.1 VARIATIONAL IMPLICIT PROCESSES FOR CLASSIFICATION

Let $\mathcal{D} \subset (\mathcal{X} \times \mathcal{Y}) = \{(\boldsymbol{x}_n, \boldsymbol{y}_n)\}_{n=1}^N$ be the training data consisting of $N$ observed pairs of input data $\boldsymbol{x}_n$ and corresponding $K$-dimensional, one-hot class label vectors $\boldsymbol{y}_n$. Deep learning approaches to classification typically assume that a class label $\boldsymbol{y}$ follows a categorical distribution whose parameters are predicted by a neural network. To this end, a network $\boldsymbol{\phi}$ with weights $\boldsymbol{w}$ defines a deterministic function $\boldsymbol{f}$ which maps an input $\boldsymbol{x} \in \mathcal{X}$ to an element $\boldsymbol{f}_{\boldsymbol{x}} \in \Delta^{K-1}$, where $\Delta^{K-1}$ denotes the $K-1$ simplex and elements from $\Delta^{K-1}$ are $K$-dimensional probability vectors. More precisely, we write $\boldsymbol{f}_{\boldsymbol{x}} = \boldsymbol{g}(\boldsymbol{x}, \boldsymbol{w}) = \boldsymbol{\sigma}(\boldsymbol{\phi}(\boldsymbol{x}; \boldsymbol{w}))$ and $\boldsymbol{y} \sim \text{Cat}(\cdot|\boldsymbol{f}_{\boldsymbol{x}})$. The softmax function $\boldsymbol{\sigma}$ ensures that $\boldsymbol{f}_{\boldsymbol{x}}$ is a valid probability vector, i.e. that $f_{\boldsymbol{x}k} \geq 0$ and $\sum_{k=1}^K f_{\boldsymbol{x}k} = 1$. In conventional ML training, the weights $\boldsymbol{w}$ are optimized by maximizing $\log p(\mathcal{D}|\boldsymbol{f}) = \log \prod_{\mathcal{D}} \text{Cat}(\boldsymbol{y}|\boldsymbol{f}_{\boldsymbol{x}}) = \sum_{\mathcal{D}} \sum_{k=1}^K y_k \log f_{\boldsymbol{x}k}$, where $\prod_{\mathcal{D}}$ and $\sum_{\mathcal{D}}$ denote $\prod_{(\boldsymbol{x},\boldsymbol{y})\in\mathcal{D}}$ and $\sum_{(\boldsymbol{x},\boldsymbol{y})\in\mathcal{D}}$ respectively, and $\boldsymbol{\phi}$, $\boldsymbol{\sigma}$, and $\boldsymbol{w}$ are implicit in $\boldsymbol{f}$.

In the Bayesian setting, this categorical predictive is expected to express epistemic uncertainty in its prediction. For many BNN models, this amounts to sampling from a weight distribution $p(\boldsymbol{w})$ and averaging the predictions to produce a posterior categorical predictive. We argue that this process *throws away* the epistemic uncertainty of the classifier, which is captured in $M$ posterior predictive samples $\boldsymbol{f}_{\boldsymbol{x}}^{(1)}, \ldots, \boldsymbol{f}_{\boldsymbol{x}}^{(M)}$. To capture this uncertainty, we study these implicit predictions and instead posit they are drawn from a posterior predictive *Dirichlet* distribution $p(\boldsymbol{f}_{\boldsymbol{x}})$. The Dirichlet density allows us to capture the variance in the categorical samples, as seen in Figure 2. In general, the model becomes a generative mapping from $\mathcal{X}$ to the space of $K$-dimensional Dirichlet distributions.

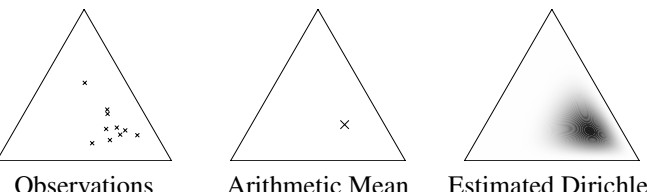

Figure 2: Reducing observations from a simplex (left) to their mean (middle) discards information which is present in the variance of the predictions that reflects model uncertainty. A Dirichlet distribution (right) fitted to the same samples can capture the uncertainty with its density function.

Sampling the weights and evaluating $\boldsymbol{g}(\boldsymbol{x}, \cdot)$ is equivalent to sampling the Dirichlet predictive, while varying $\boldsymbol{x}$ with fixed weight $\boldsymbol{w}$ corresponds to evaluating a classifier $\boldsymbol{g}(\cdot, \boldsymbol{w})$. Together, the joint variation of $\boldsymbol{w}$ and $\boldsymbol{x}$ defines an implicit stochastic process which we assume describes an implicit $K$-dimensional Dirichlet distribution for each $\boldsymbol{x} \in \mathcal{X}$. When parameterized by a neural network, we can treat this stochastic process as a variational implicit process (Example 3, Ma et al. (2019)).

**Definition 3.1** (Implicit stochastic processes, from Ma et al. (2019)). An implicit stochastic process (IP) is a set of random variables $\boldsymbol{f}$, such that any finite subset $\boldsymbol{f}_{\boldsymbol{x}_{1:L}} = \{\boldsymbol{f}_{\boldsymbol{x}_1}, \boldsymbol{f}_{\boldsymbol{x}_2}, ..., \boldsymbol{f}_{\boldsymbol{x}_L}\}$ with $L \in \mathbb{N}$ has a joint distribution which is implicitly defined by the generative process,

$$\boldsymbol{w} \sim p(\boldsymbol{w}), \qquad \boldsymbol{f}_{\boldsymbol{x}_l} = \boldsymbol{g}(\boldsymbol{x}_l, \boldsymbol{w}), \qquad \forall \boldsymbol{x}_l \in \mathcal{X}, \qquad 1 \leq l \leq L.$$

Applying Definition 3.1 to stochastic network classifiers requires two key considerations. Firstly, the weight distribution $p(\boldsymbol{w})$ takes a parameterized form, $q_{\boldsymbol{\theta}}(\boldsymbol{w})$, with parameters $\boldsymbol{\theta}$ we wish to optimize. The function $\boldsymbol{g}(\cdot, \cdot)$ represents a specific network architecture, such as a ResNet. Further details regarding the variations of $q_{\boldsymbol{\theta}}(\boldsymbol{w})$ that we consider can be found in Section D in the Appendix. Secondly, in the classification setting we are now reasoning over a joint distribution of Dirichlet distributions on a fixed $K-1$ simplex. Note this is distinct from the Dirichlet *process* (Teh, 2010), which is a trans-dimensional prior over discrete distributions, such that $K$ is inferred. Informally, in our setting, a finite collection of $L$ random variables would be an element from the $K-1$ simplex raised to the power of $L$. The stochastic process can be defined as a random variable from the $K-1$ simplex to the power of the data domain $\mathcal{X}$. This is not a distribution we can reason about easily, in contrast to the Gaussian process where the joint and marginal distributions are all Gaussian (Rasmussen & Williams, 2005). However, as we only interact with the joint distribution implicitly, we avoid the complexity of representing it directly. Using this VIP perspective, we can use the stochastic networks as an implicit variational posterior over functions $q(\boldsymbol{f}|\boldsymbol{\theta})$, allowing us to adopt high-performance BNN representations while avoiding distributions over infinite parameter spaces, as in e.g. Bayesian nonparametrics (Hjort et al., 2010). Further details regarding the formal stochastic process definition can be found in Section C in the Appendix.

While modeling the predictive distribution as a Dirichlet captures the variation in the predictive samples, the inference procedure is not affected if we use weight-space priors. To use this rich predictive density to inform our inference strategy, we can design a Dirichlet predictive prior in *function space* and apply variational function-space inference to optimize $q_{\boldsymbol{\theta}}(\boldsymbol{w})$.

## 3.2 FUNCTION-SPACE VARIATIONAL INFERENCE AND INDEX SETS

To optimize our variational implicit processes, we can use the function-space evidence lower bound objective (fELBO) (Sun et al., 2019),

$$\mathcal{L}(\boldsymbol{\theta}) = \mathbb{E}_{\boldsymbol{f} \sim q(\cdot|\boldsymbol{\theta})} [\log p(\mathcal{D}|\boldsymbol{f})] - D_{\mathrm{KL}}[q(\boldsymbol{f}|\boldsymbol{\theta}) \,\|\, p(\boldsymbol{f})], \tag{1}$$

which resembles the conventional evidence lower bound objective (ELBO) (Jordan et al., 1999; Hoffman et al., 2013). This enables the explicit specification of the desired behavior of the predictive distribution directly in function space, whereas the conventional ELBO relies on indirect specification via a prior over model parameters $p(\boldsymbol{w})$. The expected log-likelihood term in the fELBO is unchanged from the regular ELBO optimization for BNNs, where we use $M$ sample estimates using the categorical likelihood,

$$\mathbb{E}_{\boldsymbol{f} \sim q(\cdot|\boldsymbol{\theta})} [\log p(\mathcal{D}|\boldsymbol{f})] \approx \tfrac{1}{M} \sum_{n=1}^{N} \sum_{m=1}^{M} \log p\left(\boldsymbol{y}_n, \boldsymbol{f}_{\boldsymbol{x}_n}^{(m)}\right). \tag{2}$$

Since the fELBO involves distributions over functions, fVI requires computing a KL divergence between stochastic processes. Sun et al. (2019) derived this divergence as the supremum over all finite measurement sets $\boldsymbol{X} \in \mathcal{X}^L$ of index sets $\mathcal{X}^L$ of size $L$,

$$D_{\text{KL}}[q(\boldsymbol{f}|\boldsymbol{\theta}) \,||\, p(\boldsymbol{f})] = \sup_{L \in \mathbb{N}, \boldsymbol{X} \in \mathcal{X}^L} D_{\text{KL}}[q(\boldsymbol{f_X}|\boldsymbol{\theta}) \,||\, p(\boldsymbol{f_X})]. \qquad (3)$$

However this realization is impractical as this supremum is not obtainable in closed form. A tractable approximation (Sun et al., 2019; Bruinsma et al., 2021) replaces the supremum with an expectation,

$$D_{\text{KL}}[q(\boldsymbol{f}|\boldsymbol{\theta}) \,||\, p(\boldsymbol{f})] \approx \mathbb{E}_{\boldsymbol{S} \sim \mathcal{S}} \, D_{\text{KL}}[q(\boldsymbol{f_S}|\boldsymbol{\theta}) \,||\, p(\boldsymbol{f_S})] \qquad (4)$$

where $\mathcal{S} \subset \mathcal{X}^L$ is a finite set of (now) fixed size $L$, which we refer to as the 'index set'. Note that Sun et al. (2019) refer to $\mathcal{S}$ as a distribution $c$. Sun et al. (2019) motivate this approximation with the KL between two conditional stochastic processes (Section A.3, Sun et al. (2019)) which is derived for generic stochastic processes. This approximation assumes that $\mathcal{S}$ contains the points that the stochastic processes are conditioned on. For our posterior and prior process, this is the training data. Sun et al. (2019) also evaluated an adversarial approach that optimized a worst-case index set. They report that the optimized index set approximated the training data, but this index set provides inadequate regularization for out of distribution (OOD) prediction, as this predictive regularization is only applied in-distribution. The theoretical and practical requirements can be balanced by using the index set $\mathcal{S}$ sampled around the training data. Moreover, the observation that this regularization should be applied OOD suggests that the sampled index set should be drawn from the test-time data distribution. While the training data should capture this distribution, we can improve it further using unlabeled data or data augmentation. We demonstrate the benefit of this index set design in Section 4, while methods of designing effective index sets is a topic of future work.

### 3.3 POSTERIOR DIRICHLET AND KL DIVERGENCE ESTIMATION

In contrast to prior work (Joo et al., 2020; Sensoy et al., 2018; Malinin & Gales, 2018), we do not directly predict $\boldsymbol{\alpha}$ using a neural network, but instead assume $\boldsymbol{f}_{\boldsymbol{x}}^{(m)} \sim \text{Dir}(\cdot|\boldsymbol{\alpha_x})$. This allows us to compute $\boldsymbol{\alpha_x}$ through a maximum likelihood estimate (MLE) of $M$ samples $\boldsymbol{f}_{\boldsymbol{x}}^{(m)}$ from the VIP. Unfortunately, the Dirichlet MLE cannot be computed in closed form and requires fixed-point iteration or other optimization techniques to be computed, increasing the computational burden (Minka, 2000). Consequently, we replace the full MLE with a computationally cheaper alternative. To this end, we consider $\boldsymbol{\alpha_x}$ in terms of two separate but dependent parameters: the Dirichlet mean $\bar{\boldsymbol{\alpha}}_{\boldsymbol{x}} = \boldsymbol{\alpha_x}/z_{\boldsymbol{x}}$ and the Dirichlet precision $z_{\boldsymbol{x}}$. By matching the first moment of the empirical distribution of $\boldsymbol{f_x}$, we obtain an estimate of $\bar{\boldsymbol{\alpha}}_{\boldsymbol{x}} \approx \frac{1}{M} \sum_{m=1}^{M} \boldsymbol{f}_{\boldsymbol{x}}^{(m)}$. We then fix $\bar{\boldsymbol{\alpha}}_{\boldsymbol{x}}$ and estimate the precision $z_{\boldsymbol{x}}$ using a fast, iterative, quasi-Newton algorithm (Minka, 2000) using $\boldsymbol{f}_{\boldsymbol{x}}^{(1:M)} = \{\boldsymbol{f}_{\boldsymbol{x}}^{(1)}, \ldots, \boldsymbol{f}_{\boldsymbol{x}}^{(M)}\}$,

$$\left(z_{\boldsymbol{x}}^{(t+1)}\right)^{-1} = \left(z_{\boldsymbol{x}}^{(t)}\right)^{-1} + \left(z_{\boldsymbol{x}}^{(t)}\right)^{-2} \left(\frac{\mathrm{d}^2}{\mathrm{d}z_{\boldsymbol{x}}^2} \mathcal{L}_{\text{Dir}}\left(\boldsymbol{f}_{\boldsymbol{x}}^{(1:M)}, \boldsymbol{\alpha}_{\boldsymbol{x}}^{(t)}\right)\right)^{-1} \frac{\mathrm{d}}{\mathrm{d}z_{\boldsymbol{x}}} \mathcal{L}_{\text{Dir}}\left(\boldsymbol{\alpha}_{\boldsymbol{x}}^{(t)}\right), \qquad (5)$$

where $z_{\boldsymbol{x}}^{(t)}$ and $\boldsymbol{\alpha}_{\boldsymbol{x}}^{(t)} = \bar{\boldsymbol{\alpha}}_{\boldsymbol{x}}/z_{\boldsymbol{x}}^{(t)}$ are the Dirichlet precision and concentration at iteration $t$ of the quasi-Newton algorithm and $\mathcal{L}_{\text{Dir}}\left(\boldsymbol{f}_{\boldsymbol{x}}^{(1:M)}, \boldsymbol{\alpha_x}\right)$ is the Dirichlet log-likelihood $\log \prod_{m=1}^{M} \text{Dir}\left(\boldsymbol{f}_{\boldsymbol{x}}^{(m)}|\boldsymbol{\alpha_x}\right)$. This corresponds to breaking the dependence between $\bar{\boldsymbol{\alpha}}_{\boldsymbol{x}}$ and $z_{\boldsymbol{x}}$ that would be considered in the full MLE case in order to accelerate the resulting approximation. With $\bar{\boldsymbol{\alpha}}_{\boldsymbol{x}}$ and $z_{\boldsymbol{x}}$ estimated, $\boldsymbol{\alpha_x}$ and thus our variational posterior $q(\boldsymbol{f_x}|\boldsymbol{\theta}) = \text{Dir}(\boldsymbol{f_x}|\boldsymbol{\alpha_x})$ is now available. Moreover, for tractability we choose a factorized prior and assume a factorized posterior, i.e. $q(\boldsymbol{f_S}) = \prod_{l=1}^{L} q(\boldsymbol{f}_{\boldsymbol{s}_l})$, [2] so that the fKL term from Equation 3 can be estimated using $M$ samples with the Dirichlet KL divergence

$$D_{\text{KL}}[q(\boldsymbol{f_S}|\boldsymbol{\theta}) \,||\, p(\boldsymbol{f_S})] \approx \frac{1}{M} \sum_{l=1}^{L} \sum_{m=1}^{M} \left(\log q\left(\boldsymbol{f}_{\boldsymbol{s}_l}^{(m)}\Big|\boldsymbol{\theta}\right) - \log p\left(\boldsymbol{f}_{\boldsymbol{s}_l}^{(m)}\right)\right), \qquad (6)$$

where $\boldsymbol{f}_{\boldsymbol{s}}^{(m)}$ is the $m$-th implicit function evaluated at $\boldsymbol{s} \in \mathcal{S}$ and $\log q(\boldsymbol{f}_{\boldsymbol{s}}^{(m)}|\boldsymbol{\theta})$ and $\log p(\boldsymbol{f}_{\boldsymbol{s}}^{(m)})$ are the log-likelihood of $\boldsymbol{f}_{\boldsymbol{s}}^{(m)}$ under the variational Dirichlet posterior and Dirichlet prior respectively. Note, in this work our predictive prior $p(\boldsymbol{f_x}) = \text{Dir}(\cdot \,|\, \boldsymbol{\beta})$, where each $\beta_k$ is chosen to be 1 unless

---

[2] While this is not a rich prior, we are assuming our VIP, e.g. CNNs, will capture meaningful correlations.

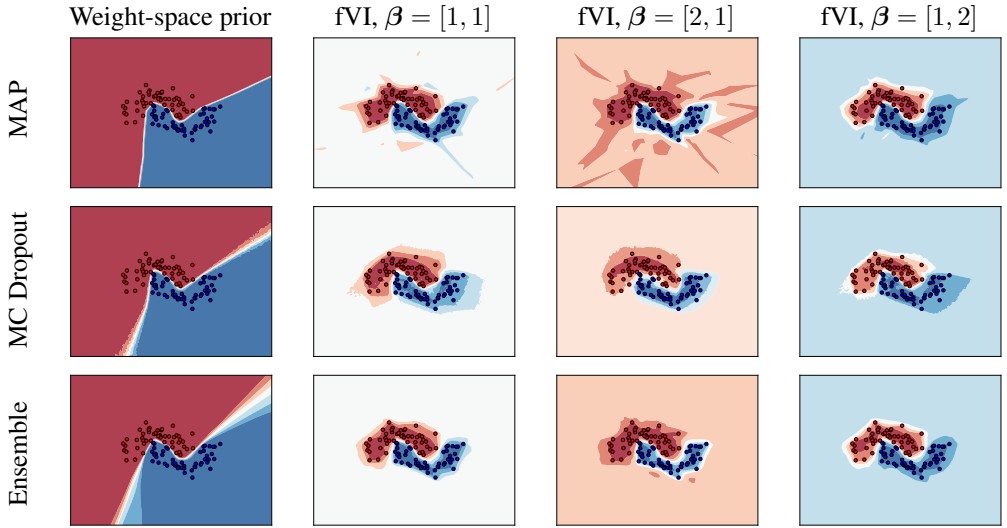

Figure 3: Toy classification problem using the Two Moons dataset. Each row corresponds to a different network. The first column shows the undesirable predictive certainty of standard weight-space priors outside of the data distribution. The second column illustrates how our function-space inference approach combined with a uniform Dirichlet prior prevents unreasonable extrapolation behavior and instead adequately increases model uncertainty outside of the observed data. The third and fourth column show that our approach can also be combined with class-biased predictive priors that inform the OOD-prediction without sacrificing accuracy. This explicit design is not straightforward with weight-space priors.

otherwise stated. During optimization, we treat $q(\boldsymbol{f_x}|\boldsymbol{\theta})$ as constant with respect to $\boldsymbol{\theta}$ due to the $\boldsymbol{\alpha_x}$ estimate, which serves the practical purpose of pruning the Dirichlet MLE from the computation graph and evokes expectation maximization-style inference. While this simplification removes the influence of $\boldsymbol{\theta}$ on $\mathrm{Dir}(\boldsymbol{f_x}|\boldsymbol{\alpha_x})$ on the KL in the gradient update, learning is not hindered due to the iterative nature of SGD training in practice. From the perspective of the wake-sleep inference procedure adopted by Ma et al. (2019), optimizing the fELBO represents the wake phase and the Dirichlet estimation represents the sleep phase. Nonetheless, we optimize $\boldsymbol{\theta}$ using backpropagation on the fELBO objective by optimizing through the function samples w.r.t. the expected log-likelihood and fixed posterior and prior densities in the fKL approximation. For some models (e.g. ensembles) this is standard gradient descent, while for others (e.g. MFVI) the reparameterization trick is required. In the case where only a single sample is available ($M = 1$), we approximate the precision $z_{\boldsymbol{x}}$ with the size of the training data. Further practical considerations, mini-batching and prior regularization, are discussed in Section B in the Appendix. Both considerations manifest as a scaling of the KL term.

## 4 EXPERIMENTS

In this section, we present an empirical evaluation of our proposed fVI approach on classification tasks for several models, comparing performance against their conventional training procedure. We first visualize the effects of a Dirichlet prior on a toy problem, and then perform image classification on the MNIST (LeCun et al., 2010), CIFAR10 and CIFAR100 (Krizhevsky, 2009) data sets. Depending on the experiment, we either employ feedforward multilayer perceptrons (MLPs) or convolutional neural networks (CNNs). Metrics include classification accuracy, log-likelihood (LLH) and expected calibration error (ECE) (Naeini et al., 2015), which estimates the calibration of accuracy vs confidence through binning the predicted class probabilities. Additional details are provided in Appendix F.

### 4.1 TOY PROBLEM

To visualize the effects of function-space variational inference, we conducted a toy experiment with the Two Moons data set. We used three different model archetypes in this experiment: MAP (i.e. a deterministic baseline), MC Dropout (Gal & Ghahramani, 2016), and (deep) Ensembles (Lakshminarayanan et al., 2017). Figure 3 shows that our inference approach combined with a uniform Dirichlet prior adequately increases predictive uncertainty outside of the observed data, whereas conventional weight-space training leads to overconfident extrapolation. While the weight-

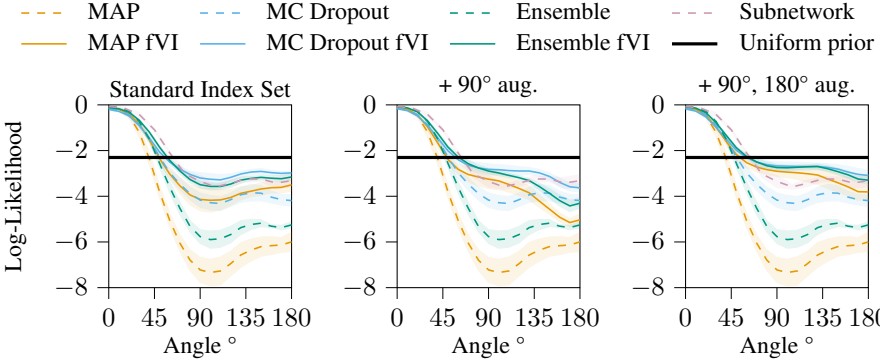

Figure 4: Comparison of rotated MNIST log-likelihood for models trained with fVI using different index sets: training data without rotations (left), 90° augmentations (middle), and 90° and 180° augmentations. Colored lines denote the mean and shaded areas denote two standard deviations over 10 seeds. The 90° augmentation improved fVI performance for that scale of perturbation, but worsens performance at 180°. We attribute the degradation to the nature of parametric function approximation. Augmenting the index set further with 180° transforms fixes the degradation, demonstrating an overall improvement in UQ over the initial training data index set. The horizontal black line denotes the performance of the uniform prior. The subnetwork performance is taken from Daxberger et al. (2021), and uses a ResNet-18 rather than an MLP. Further metrics are reported in Figure 9.

space approach learns a decision boundary that bisects the data, the function-space approach learns richer decision boundaries that more accurately capture the data distribution. This behavior has also been observed with Gaussian process-based classification, e.g. Liu et al. (2020). Furthermore, fVI facilitates specifying class-biased priors, which would be difficult to achieve with weight priors.

## 4.2 ROTATED MNIST

Following Ovadia et al. (2019), we train on the MNIST handwritten digit classification dataset (LeCun et al., 2010) and evaluate on constructed test data with rotations of up to 180°, which simulates a challenging OOD scenario due to the absence of data augmentation. For this experiment, we used the same model archetypes and MLP architecture as for the toy problem in Section 4.1.

The log-likelihood between models is shown in Figure 4 (left), accuracy and expected calibration error are reported in Figure 9 in the Appendix. In terms of classification error rate, both approaches yield the same performance. In terms of log-likelihood, fVI consistently outperforms their weight-space counterparts as the data becomes more OOD, indicating better OOD uncertainty quantification. Subnetwork linearized Laplace (Daxberger et al., 2021) is also reported as a competitive baseline, achieving better accuracy and in-distribution log-likelihood. However, these results were obtained using a ResNet-18 architecture rather than an MLP.

## 4.3 ASSESSING INDEX SET DESIGN

To illustrate the importance of the index set, we train the fVI models for rotated MNIST using three different index sets: the training data, additional 90° augmentation, and additional 90° and 180° augmentation. While simply using the training data without rotations as index set for fVI already outperforms the weight-space counterparts, a direct comparison in Figure 4 illustrates that performance can be further increased if an appropriate index set, i.e. example OOD data, is available. With the enriched index sets, the OOD performance move closer to that of the prior, indicating more accurate inclusion in the fELBO. Sets for greater OOD performance could be designed through manual data augmentation, unlabeled data or synthetic data generation. Note, for all other image classification experiments, we use the training data as the index set, demonstrating performance without this inductive bias.

## 4.4 LARGER-SCALE IMAGE CLASSIFICATION UNDER CORRUPTION

We used the regular train splits of the CIFAR10 and CIFAR100 (Krizhevsky, 2009) image classification data sets as training data and their corrupted versions (Hendrycks & Dietterich, 2019) as OOD test data. CIFAR10 and CIFAR100 consist of natural color images of animals and vehicles, and the corrupted versions perturb the images at 5 increasing levels of severity by changing the brightness,

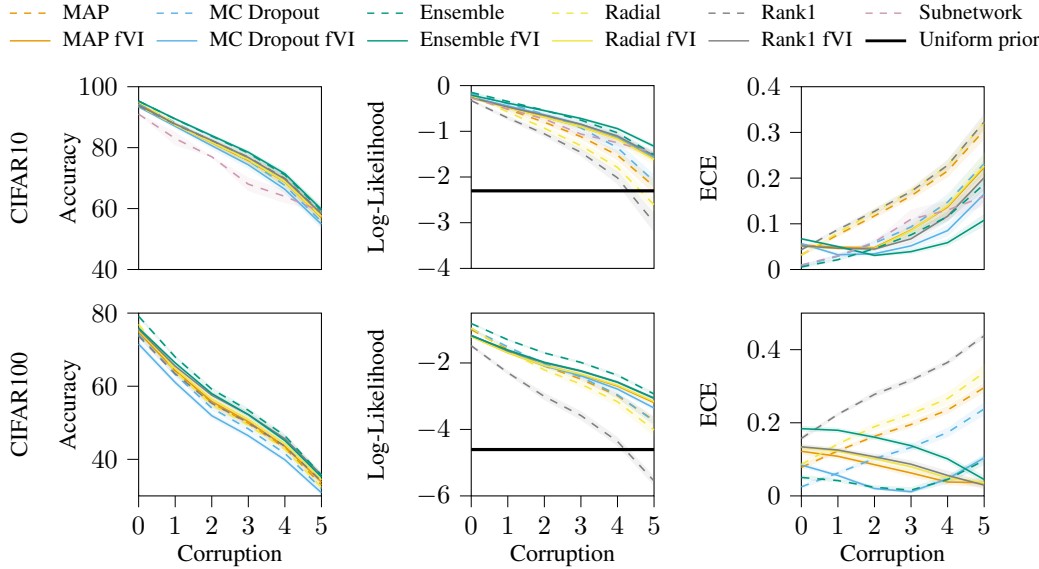

Figure 5: Metrics for corrupted image classification on CIFAR10 (top) and CIFAR100 (bottom). All models use a ResNet-18 architecture. For CIFAR10, there is a clear benefit of fVI priors over weight-space for log-likelihood. Accuracy performance is constant, while fVI underfits for ECE when in-distribution due to regularization from the prior. For CIFAR100, the higher label dimensionality results in stronger regularization from the uniform prior. As a result, weight-space ensembles achieve better performance over all corruptions. Moreover, fVI models underfit w.r.t. ECE in-distribution due to the prior, but improve in calibration OOD, whereas weight-space models worsen. This result indicates prior specification requires more care for high-dimensional classification. Interestingly, the fVI performance is consistent across models for accuracy and log-likelihood on both datasets, but are diverse for ECE. The linearized Laplace subnetwork result is taken from Daxberger et al. (2021).

contrast and saturation, or adding noise, blur or other artifacts, such that it becomes ever more difficult for the models to classify them. For this experiment, we used CNN models based on the ResNet-18 architecture (He et al., 2016). In addition to the previously used MAP, MC Dropout and Ensemble models, we also evaluate our fVI objective on Radial BNNs (Farquhar et al., 2020), as an effective variant of MFVI, and Rank1 BNNs (Dusenberry et al., 2020), which combine ensembles and VI.

Figure 5 reports the performance metrics for CIFAR10 and CIFAR100 under corruption. Tabulated results can be found in Section G in the Appendix. The general trend of these results is that the function-space prior frequently provides gains in OOD UQ with only a small decrease in (uncorrupted) test performance. This trade-off between accuracy and robustness has been observed and discussed in the adversarial robustness setting (Tsipras et al., 2019; Yang et al., 2020), and remains an open problem if and how both qualities can be achieved in practice. Moreover, the shared function-space prior resulted in remarkable consistency across models, compared to the variety seen in weight-space priors. Due to the predictive prior, the ECE was significantly higher for corruption 0, but this underfitting provided superior ECE when OOD. This suggests the function-space prior should be fine-tuned (i.e. empirical Bayes) if superior ECE is desired in-distribution. For CIFAR100, the higher prior regularization due to higher dimensionality (see Section B.2), resulted in reduced benefit over weight-space models compared to CIFAR10, with improved performance only evident at stronger corruptions. Figure 7 summarizes the LLH difference due to the function-space prior.

## 4.5 LARGER-SCALE IMAGE CLASSIFICATION UNDER ADVERSARIAL ATTACKS

Despite the success of CNNs in computer vision, adversarial attacks are one of the biggest risks when it comes to practical applications (Akhtar & Mian, 2018). We evaluate the robustness of fVI compared to standard weight-prior approaches on the CIFAR10 and CIFAR100 datasets using the fast gradient sign method (FGSM) (Goodfellow et al., 2014). Figure 6 compares the accuracy and the log-likelihood of the test data with increasing amounts of perturbation, ranging from $\epsilon = 0$ (no attack) to $\epsilon = 0.3$. Although both weight-space and function-space models lose their classification accuracy when the FGSM attack is introduced, the fVI models only suffer small decreases in log-likelihood, whereas the weight-space LLH performance drops significantly. We also observe the accuracy vs

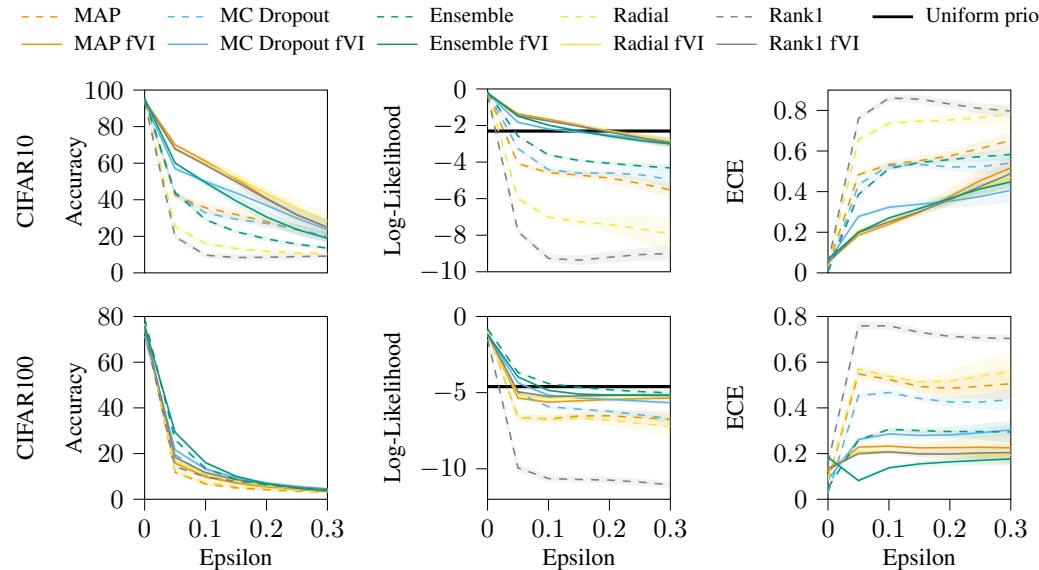

Figure 6: Metrics for adversarial examples on CIFAR10 (top) and CIFAR100 (bottom). All models use a ResNet-18 architecture. For CIFAR10, there are significant benefits of fVI over weight-space approaches across all metrics. For CIFAR100, the fVI benefits are still evident, but the higher label dimensionality results in stronger regularization from the uniform prior. As a result, the weight-space ensembles achieve slightly better performance over all epsilons.

robustness trade-off in the fVI models discussed in Section 4.4. We attribute this behavior to the quality of the uncertainty quantification at the decision boundary. While both approaches have brittle boundaries due to the nature of CNNs, the predictive uncertainty at these decision boundaries is richer for fVI. This richness is illustrated in the toy example in Figure 3.

## 5 CONCLUSION

We propose a function-space variational inference approach to deep Bayesian classification, which enables the use of Dirichlet predictive priors to improve uncertainty quantification. Our approach provides a unified formalism of prior work on Dirichlet-based classifiers with function-space regularization, and can be applied to a general class of variational implicit processes without altering their underlying architectures. Experiments demonstrate that our approach generally outperforms the corresponding weight-space priors in terms of uncertainty quantification and adversarial robustness. By considering the design of the index set, we show how one can trade-off scalability against OOD uncertainty quantification by extending the fKL evaluation beyond the training distribution. Future research should develop index sets for fVI, specifically effective methods for constructing them to reflect the test distribution, e.g. through using data augmentation or unlabeled data.

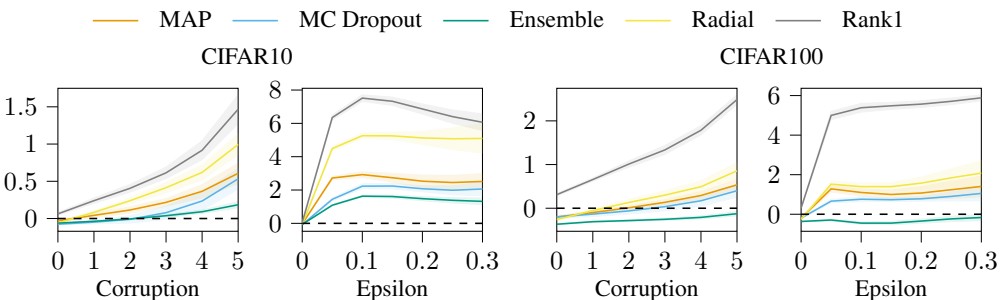

Figure 7: A summary of the improvement in log-likelihood for function-space inference over weight space, for CIFAR10 and CIFAR100 experiments under corruption and adversarial perturbations. The general trend shows that models that use reparameterization gradients (Radial and Rank1) benefit the most from fVI. The improvement also generally increases as the test data becomes more OOD. Ensembles are improved the least, and even weight-space inference outperforms function-space for CIFAR100. This result is attributed to the regularization of the uniform prior in high dimensions.

## ETHICS STATEMENT

As this paper looks at the abstract problem of classification under a Bayesian lens, we are not aware of direct ethical implications of this work.

## REPRODUCIBILITY STATEMENT

Code for reproducing experiment results is provided in the supplementary material. Further, Appendices F, G and H contain additional details and results of the experiments.

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

## A    RELATION TO PRIOR WORK

In this section, we discuss similarities and differences to prior work with Dirichlet posteriors in more detail by comparing assumptions and optimization objectives. To this end, we first explain prior work using their own notation and then discuss the correspondence to our fVI perspective.

### A.1    PRIOR NETWORKS

Malinin & Gales (2018; 2019) use a neural network with parameters $\boldsymbol{\theta}$ to directly predict the concentration parameters $\alpha_c$ of a Dirichlet distribution $\mathrm{p}(\boldsymbol{\mu}|\boldsymbol{x};\boldsymbol{\theta})$, given input $\boldsymbol{x}$, which is distinct from our approach of estimating a posterior Dirichlet from $M$ categorical predictions. This model is not a Bayesian network in practice, as only point estimates for the weights are learned. To ensure $\alpha_c > 0$, an element-wise exponential operation is applied as the final layer of the neural network. Additionally, prior networks minimize an optimization objective consisting of two separate KL divergences, representing in- and out-of-distribution data respectively,

$$\mathcal{L}(\boldsymbol{\theta}) = \mathbb{E}_{\mathrm{p_{in}}(\boldsymbol{x})}[D_{\mathrm{KL}}[\mathrm{Dir}(\boldsymbol{\mu}|\hat{\boldsymbol{\alpha}}) \| \mathrm{p}(\boldsymbol{\mu}|\boldsymbol{x};\boldsymbol{\theta})]] + \mathbb{E}_{\mathrm{p_{out}}(\boldsymbol{x})}[D_{\mathrm{KL}}[\mathrm{Dir}(\boldsymbol{\mu}|\tilde{\boldsymbol{\alpha}}) \| \mathrm{p}(\boldsymbol{\mu}|\boldsymbol{x};\boldsymbol{\theta})]]. \quad (7)$$

The first expectation $\mathbb{E}_{\mathrm{p_{in}}}$ accounts for the actual learning, i.e. fitting the training data, whereas the second expectation $\mathbb{E}_{\mathrm{p_{out}}}$ is supposed to regularize the model by matching a prior distribution. Accordingly, the first expectation is computed for the training data and can be compared to the expected log-likelihood term in our fVI approach.  Instead of maximizing the categorical log-likelihood of $M$ observations, prior networks construct Dirichlet targets by smoothing categorical ground truth labels to define the Dirichlet mean and setting the precision as a hyperparameter during training. Although we also apply 'label smoothing' to the predictions, it is for numerical reasons and not for the construction of target distributions from labels. Additionally, prior networks treat the precision of their constructed *target* distribution as a hyperparameter, whereas we estimate the Dirichlet precision of our *predicted* variational posterior distribution via maximum likelihood. The second expectation is computed for OOD data and resembles the fKL term in our fVI approach, where the OOD data is used as the index set. In contrast to prior networks, the more general fKL formulation also allows the training data or mixtures of training data and OOD data as index sets, whereas prior networks explicitly compute their second expectation for OOD data only. Furthermore, both prior network expectations consider the KL divergence from the neural network predictive distribution (right) to the target or prior distribution (left), whereas, in our fVI approach, and variational inference in general, the KL divergence from the prior distribution (right) to the variational posterior (left) is considered.

### A.2    BELIEF MATCHING

Joo et al. (2020) assumes a Dirichlet prior which, together with the categorical ground truth class labels, define a target Dirichlet posterior. A neural network is used to directly predict concentration parameters of a Dirichlet posterior $q_{\boldsymbol{z}|\boldsymbol{x}}$ by replacing the final softmax layer with an element-wise exponential operation. To learn the target posterior, belief matching maximizes

$$l_{EB}(\boldsymbol{y}, \alpha^{\boldsymbol{W}}(\boldsymbol{x})) = \mathbb{E}_{q_{\boldsymbol{z}|\boldsymbol{x}}}[\log \boldsymbol{z_y}] - D_{\mathrm{KL}}[q_{\boldsymbol{z}|\boldsymbol{x}}^{\boldsymbol{W}} \| p_{\boldsymbol{z}|\boldsymbol{x}}], \quad (8)$$

where $\mathbb{E}_{q_{\boldsymbol{z}|\boldsymbol{x}}}[\log \boldsymbol{z_y}]$ is the expected log-likelihood of the training data and $D_{\mathrm{KL}}q_{\boldsymbol{z}|\boldsymbol{x}}^{\boldsymbol{W}}p_{\boldsymbol{z}|\boldsymbol{x}}$ is the KL divergence between the predicted Dirichlet posterior and the Dirichlet prior. Therefore, their objective matches our fELBO objective equation 1 except for two differences: Firstly, belief matching computes both the expected log-likelihood and the KL divergence with respect to their single, directly predicted Dirichlet distribution, whereas we evaluate them as arithmetic averages of $M$ stochastic categorical model outputs. Secondly, belief matching does not recognize the function-space aspect and instead only considers evaluation of the KL divergence using the training data, which resembles our fKL where the index set is constrained to be the training data.

### A.3    EVIDENTIAL DEEP LEARNING

Sensoy et al. (2018) directly predicts the concentration parameter of a Dirichlet distribution by using a neural network with ReLU activations as final layer to assert the positive constraint. Additionally, a loss function is derived via type-II maximum likelihood by integrating over a Dirichlet prior and the

sum of squares between target labels $\boldsymbol{y}_i$ and predicted probabilities $\boldsymbol{p}_i$. Furthermore, a regularizing KL divergence term is added, resulting in a total loss function,

$$\mathcal{L}(\Theta) = \sum_{i=1}^{N} \left( \sum_{j=1}^{K} \left( (y_{ij} - \hat{p}_{ij})^2 + \frac{\hat{p}_{ij}(1 - \hat{p}_{ij})}{S_i + 1} \right) + \lambda_t D_{\mathrm{KL}}[\mathrm{Dir}(\boldsymbol{p}_i | \tilde{\boldsymbol{\alpha}}_i) \,||\, \mathrm{Dir}(\boldsymbol{p}_i | \mathbf{1})] \right), \quad (9)$$

where $y_{ij}$ are individual 0-1 target labels, $\hat{p}_{ij}$ are components of the predicted Dirichlet mean, $S_i$ is the predicted Dirichlet precision, $\tilde{\boldsymbol{\alpha}}_i$ is the predicted Dirichlet concentration parameter, $\mathbf{1}$ is a vector of ones and $\lambda_t$ is a annealing coefficient for optimization. The first part of their loss is responsible for fitting the training data and can thus be compared to the maximum likelihood objective in Section 3.1. The ML objective can be derived from the categorical log-likelihood via type-I maximum likelihood, whereas their objective is derived by minimizing the sum of squares via type-II maximum likelihood. The second part of their loss resembles the fKL in our fVI approach. However, they only evaluate the KL divergence for the training data and explicitly consider the uniform Dirichlet distribution with concentration $\mathbf{1}$. Therefore, their KL divergence regularization term is a special case of our fKL with the index set being the training data and the prior being the uniform Dirichlet distribution. For both parts, a major difference between evidential deep learning and our approach is the realization of the predictive Dirichlet distribution. Evidential deep learning directly predicts Dirichlet concentration parameters, whereas we use $M$ categorical predictions to estimate a Dirichlet distribution via maximum likelihood.

### A.4 EXPERIMENTAL COMPARISON

In this subsection, we compare our MAP and MAP fVI models to this related work from Appendix A, namely Belief Matching and Prior Networks, which both demonstrated scalability to ResNet models. To reproduce their results, we used the official open-source implementations [3] [4].

Figure 8 illustrates the test accuracy, log-likelihood, and expected calibration error for the corrupted CIFAR10 image classification task (see Section 4.4). We trained the models using the same procedure and hyperparameters described in Section F. The Belief Matching model corresponds closely to the MAP fVI model, as both the objectives and models are similar. Unfortunately, we were not able to reproduce the Prior Networks performance described in the paper (Malinin & Gales, 2019), neither with their listed hyperparameters (Table 4, (Malinin & Gales, 2019)) or the hyperparameters used in Section F. It is uncertain whether this is due to the model, implementation bugs or unrecorded hyperparameters [5].

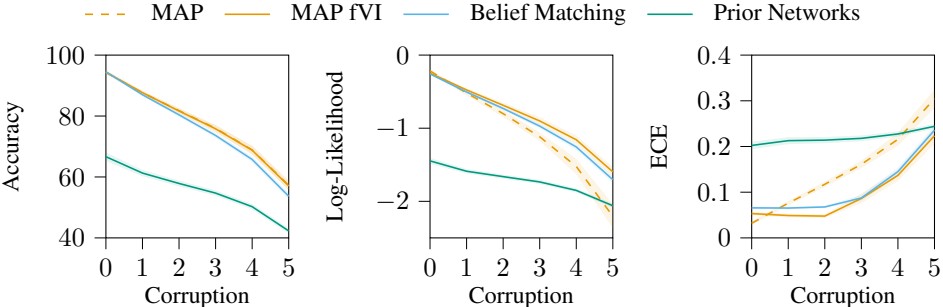

Figure 8: Comparing Prior Networks and Belief Matching against MAP and MAP fVI models on the corrupted CIFAR10 task.

---

[3] github.com/tjoo512/belief-matching-framework
[4] github.com/KaosEngineer/PriorNetworks
[5] The authors did not respond to personal correspondence regarding this matter.

# B PRACTICAL CONSIDERATIONS

This section discusses practical issues with the approach in more detail.

## B.1 MINI-BATCH TRAINING

In weight-space variational inference, batch training requires an appropriate scaling term to the KL divergence of the ELBO during each batch iteration. This is to ensure the weight KL is correctly balanced against the batch expected log-likelihood. The fKL approximation is equally mini-batched, as we restrained into using a small portion of the index set each iteration. Therefore, like Sun et al. (2019), we divide both the batched expected log-likelihood and the fKL by the mini-batch size $B$ for numerical stability. The result of this is that the fKL weight in the total ELBO then depends on the mini-batch size, which is theoretically undesirable. However, in practice the KL divergence scaling in variational inference is a topic of active debate (Wenzel et al., 2020; Aitchison, 2021) and is frequently scaled or annealed for numerical reasons (e.g. Dusenberry et al. (2020)).

## B.2 PRIOR SPECIFICATION FOR HIGH OUTPUT DIMENSION

When scaling to higher dimensional classification tasks, specifically $K \geq 100$, we observed numerical issues with the fELBO objective when using the uniform Dirichlet predictive prior. In higher dimensions, this prior would provide greater regularization. This is because the magnitude of the categorical likelihood does not change with dimensionality, as it is the log probability of the label class. Conversely, the KL between two Dirichlet densities requires summing over the parameters, so the magnitude naturally increases with $K$. To alleviate this over-regularization, we adopt the strategy of Joo et al. (2020) and apply additional scaling to the KL term in the fELBO. This scaling can be shown to be numerically equivalent to a certain prior, i.e. $\beta$ (Section 3.4, Joo et al. (2020)). Therefore, optimizing this scaling is a form of model selection. For our CIFAR100 experiments, we simply chose a scaling such that that the fKL magnitude was close to the CIFAR10 values. We found this to be about $0.1$, which matches a 10x scaling suggested by the Dirichlet KL due to the summation terms.

## C    DEEP STOCHASTIC CLASSIFIERS AS STOCHASTIC PROCESSES

Stochastic classifiers are stochastic processes instantiated as a feedforward or convolutional neural networks with stochastic weights. Formally, our stochastic process $\boldsymbol{f}$ is defined on the sample space $\Omega$ with an index set $\mathcal{X}$ defined by the data type (e.g. real vectors, RGB images), so $\boldsymbol{f} : \mathcal{X} \times \Omega \to \Delta^{K-1}$, where $\Delta^{K-1}$ is the state space, which is the $K-1$ simplex. A random variable $\boldsymbol{f}(\boldsymbol{x}) : \Omega \to \Delta^{K-1}$ can be defined for each $\boldsymbol{x} \in \mathcal{X}$ and we write $\boldsymbol{f}(\boldsymbol{x}) = \boldsymbol{f_x}$.

Kolmogorov's extension theorem (Tao, 2013) guarantees the existence of a stochastic process $\boldsymbol{f}$ if for each $L \in \mathbb{N}$ the finite marginal joint distributions $p_{1:L}(\boldsymbol{f_{x_{1:L}}})$, where $\boldsymbol{f_{x_{1:L}}} = \{\boldsymbol{f_{x_1}}, \ldots, \boldsymbol{f_{x_L}}\}$, satisfy exchangeability and consistency. These finite marginal joint distributions are implicitly defined according to Definition 3.1.

**Exchangeability**: For any permutation $\pi$ of $1, \ldots, L$, $p_{\pi(\boldsymbol{x}_{1:L})}(\boldsymbol{f}_{\pi(x_{1:L})}) = p_{\boldsymbol{x}_{1:L}}(\boldsymbol{f_{x_{1:L}}})$. This requires that the process behavior is invariant to the order of inputs. For a feedforward neural network, this is satisfied because the respective predictions do not change if the order of inputs changes.

**Consistency**: For any $1 \leq L' < L$, $p_{1:L'}(\boldsymbol{f_{x_{1:L'}}}) = \int p_{1:L}(\boldsymbol{f_{x_{1:L}}}) \, \mathrm{d}\boldsymbol{f_{x_{L'+1:L}}}$. This requires that future evaluations are independent of past evaluations. For a feedforward neural network, this is satisfied because predictions do not depend on previous predictions.

## D    VARIATIONAL IMPLICIT PROCESS PARAMETERIZATIONS

This work uses popular classes of deep stochastic models used for weight-space BNNs, and interprets them as variational implicit processes in order to train them with function-space inference. To clarify this connection, Table 1 summarizes the connection between popular BNN models (and the MAP baseline) and weight distribution $q_{\boldsymbol{\theta}}(\boldsymbol{w})$.

Table 1: Summary of stochastic (and deterministic) representations for the network weights $\boldsymbol{w}$, which correspond to popular BNN approaches. These distributions apply either to the whole network or per layer. Note $\boldsymbol{\Sigma}_\theta$ is typically factorized in practice. While we use Radial BNNs rather than MFVI, we include it for completeness. Rank1 references specifically the Gaussian realization.

| Model | Parameterization for $\boldsymbol{w} \sim q_{\boldsymbol{\theta}}(\cdot)$ | Scope |
|---|---|---|
| MAP | $\delta(\boldsymbol{w} - \boldsymbol{w_\theta})$ | Network |
| MFVI | $\mathcal{N}(\boldsymbol{\mu_\theta}, \boldsymbol{\Sigma_\theta})$ | Layer |
| Radial | $\boldsymbol{w} \sim \boldsymbol{\mu_\theta} + \sqrt{\boldsymbol{\Sigma}_\theta} \odot \hat{\boldsymbol{\epsilon}} \cdot |r|, \ \hat{\boldsymbol{\epsilon}} = \boldsymbol{\epsilon}/|\boldsymbol{\epsilon}|, \ \boldsymbol{\epsilon} \sim \mathcal{N}(\boldsymbol{0}, \boldsymbol{I}), \ r \sim \mathcal{N}(0, 1)$ | Layer |
| MC Dropout | $\boldsymbol{w} \sim \bar{\boldsymbol{w}}_{\boldsymbol{\theta}} \odot \boldsymbol{b}, \ b_i \sim \text{Bernoulli}(p_{\text{dropout}})$ | Layer |
| Ensemble | $\frac{1}{M} \sum_m \delta(\boldsymbol{w} - \boldsymbol{w}_{\boldsymbol{\theta}}^m)$ | Network |
| Rank1 | $\frac{1}{M} \sum_m \delta(\boldsymbol{w} - \boldsymbol{w}_{\boldsymbol{\theta}}^m), \ \boldsymbol{w}_{\boldsymbol{\theta}}^m \sim \bar{\boldsymbol{w}}_{\boldsymbol{\theta}} \odot \delta\boldsymbol{w}_{\boldsymbol{\theta}}^m, \ \delta\boldsymbol{w}_{\boldsymbol{\theta}}^m \sim \mathcal{N}(\boldsymbol{\mu}_{\boldsymbol{\theta}}^m, \boldsymbol{v}_{\boldsymbol{\theta}}^m \boldsymbol{u}_{\boldsymbol{\theta}}^{m\top})$ | Layer |

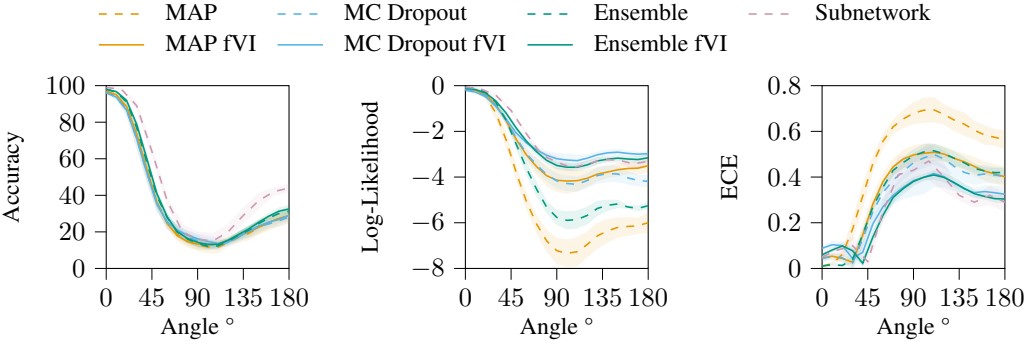

Figure 9: Metrics for rotated MNIST. All models use a MLP architecture with two hidden layers with 50 units each. In terms of log-likelihood and expected calibration error, the fVI models outperform their respective baselines, which indicates improved uncertainty quantification. The linearized laplace subnetwork metrics are taken from Daxberger et al. (2021), and uses a ResNet-18 rather than an MLP.

# E  DIRICHLET MUTUAL INFORMATION

Smith & Gal (2018) proposed to use the mutual information $\mathcal{MI}(\boldsymbol{\theta}; \boldsymbol{y})$ of model parameters $\boldsymbol{\theta}$ and class label $\boldsymbol{y}$ to detect adversarial examples. Intuitively, this quantity measures how much the model could improve its parameters $\boldsymbol{\theta}$ if $\boldsymbol{y}$ were observed. Therefore, it quantifies epistemic uncertainty and is also known as information gain. However, Smith & Gal (2018) note that, in general, $\mathcal{MI}(\boldsymbol{\theta}; \boldsymbol{y})$ is intractable for deep learning models. Therefore, they provide a tractable approximation which applies to models with stochastic parameters and requires sampling-based estimation.

Fortunately, in our function-space variational setting, where the model parameters $\boldsymbol{\theta}$ are deterministic, the one-hot encoded class label $\boldsymbol{y}$ follows a categorical distribution $\boldsymbol{y} \sim \mathrm{Cat}\left(\cdot|\boldsymbol{f_x}\right)$ and the latent probability vector $\boldsymbol{f_x}$ follows a Dirichlet distribution $\boldsymbol{f_x} \sim \mathrm{Dir}\left(\cdot|\boldsymbol{\alpha}\right)$, the equivalent mutual information between class labels and model predictions $\mathcal{MI}\left(\boldsymbol{f_x}; \boldsymbol{y}\right) = \mathcal{H}\left(\boldsymbol{y}\right) - \mathcal{H}\left(\boldsymbol{y}|\boldsymbol{f_x}\right)$ can be derived in closed-form from the Dirichlet approximation using

$$\mathcal{H}\left(\boldsymbol{y}\right) = -\sum_{k=1}^{K} \frac{\alpha_k}{z} \log\left(\frac{\alpha_k}{z}\right), \tag{10}$$

$$\mathcal{H}\left(\boldsymbol{y}|\boldsymbol{f_x}\right) = -\sum_{k=1}^{K} \frac{\alpha_k}{z}\left(\psi\left(\alpha_k + 1\right) - \psi\left(z + 1\right)\right). \tag{11}$$

The complete derivation is provided in Appendix E.1 on the next page.

The first term $\mathcal{H}\left(\boldsymbol{y}\right)$ corresponds to the categorical entropy of the expected value of $\boldsymbol{f_x}$. It is equivalent to the standard measure of predictive entropy for softmax-based ML deep learning classifiers. The second term $\mathcal{H}\left(\boldsymbol{y}|\boldsymbol{f_x}\right)$ is the conditional entropy of the class label $\boldsymbol{y}$ given a Dirichlet observation $\boldsymbol{f_x}$ and it is influenced by the spread of $\boldsymbol{f_x}$, whereas the first term only depends on its mean. Although this function-space variational formulation admits closed-form solutions for $\mathcal{MI}\left(\boldsymbol{f_x}; \boldsymbol{y}\right)$, the resulting values will still be approximations in our setting because the variational posterior $\mathrm{Dir}\left(\boldsymbol{f_x}|\boldsymbol{\alpha_x}\right)$ was estimated from $M$ observations. Nonetheless, it is not limited to models with stochastic weights, but instead provides a unified approach to measuring epistemic uncertainty for any classifier with a Dirichlet predictive distribution.

## E.1 Mutual Information Derivation

Let $\boldsymbol{y} \sim \operatorname{Cat}\left(\cdot | \boldsymbol{f_x}\right)$, $\boldsymbol{f_x} \sim \operatorname{Dir}\left(\cdot | \boldsymbol{\alpha}\right)$, $f_{\boldsymbol{x}k} \sim \operatorname{Beta}\left(\cdot | a_k, b_k\right)$, where $a_k = \alpha_k$ and $a'_k = a_k + 1$, while $b_k = z - \alpha_k$ and $z = \sum_{k=1}^K \alpha_k$,

$$\mathcal{MI}\left(\boldsymbol{f_x}; \boldsymbol{y}\right) = \mathcal{H}\left(\boldsymbol{y}\right) - \mathcal{H}\left(\boldsymbol{y} | \boldsymbol{f_x}\right), \tag{12}$$

$$\mathcal{H}\left(\boldsymbol{y}\right) = -\sum_{k=1}^K \mathbb{E}_{\boldsymbol{f_x}}\left[f_{\boldsymbol{x}k}\right] \log\left(\mathbb{E}_{\boldsymbol{f_x}}\left[f_{\boldsymbol{x}k}\right]\right), \tag{13}$$

$$= -\sum_{k=1}^K \mathbb{E}_{f_{\boldsymbol{x}k}}\left[f_{\boldsymbol{x}k}\right] \log\left(\mathbb{E}_{f_{\boldsymbol{x}k}}\left[f_{\boldsymbol{x}k}\right]\right), \tag{14}$$

$$= -\sum_{k=1}^K \frac{\alpha_k}{z} \log\left(\frac{\alpha_k}{z}\right), \tag{15}$$

$$\mathcal{H}\left(\boldsymbol{y} | \boldsymbol{f_x}\right) = \mathbb{E}_{\boldsymbol{f_x}}\left[-\sum_{k=1}^K f_{\boldsymbol{x}k} \log\left(f_{\boldsymbol{x}k}\right)\right], \tag{16}$$

$$= -\sum_{k=1}^K \mathbb{E}_{\boldsymbol{f_x}}\left[f_{\boldsymbol{x}k} \log\left(f_{\boldsymbol{x}k}\right)\right], \tag{17}$$

$$= -\sum_{k=1}^K \mathbb{E}_{f_{\boldsymbol{x}k}}\left[f_{\boldsymbol{x}k} \log\left(f_{\boldsymbol{x}k}\right)\right], \tag{18}$$

$$= -\sum_{k=1}^K \int_0^1 f_{\boldsymbol{x}k} \log(f_{\boldsymbol{x}k}) \frac{f_{\boldsymbol{x}k}^{a_k-1}(1 - f_{\boldsymbol{x}k})^{b_k-1}}{\mathrm{B}(a_k, b_k)} \mathrm{d}f_{\boldsymbol{x}k}, \tag{19}$$

$$= -\sum_{k=1}^K \int_0^1 \log(f_{\boldsymbol{x}k}) \frac{f_{\boldsymbol{x}k}^{a_k}(1 - f_{\boldsymbol{x}k})^{b_k-1}}{\mathrm{B}(a_k, b_k)} \mathrm{d}f_{\boldsymbol{x}k}, \tag{20}$$

$$= -\sum_{k=1}^K \int_0^1 \log(f_{\boldsymbol{x}k}) \frac{f_{\boldsymbol{x}k}^{a'_k-1}(1 - f_{\boldsymbol{x}k})^{b_k-1}}{\mathrm{B}(a_k, b_k)} \mathrm{d}f_{\boldsymbol{x}k}, \qquad \lim_{c \to 0} \frac{x^c - 1}{c} = \log x \tag{21}$$

$$= -\sum_{k=1}^K \frac{1}{\mathrm{B}(a_k, b_k)} \int_0^1 \frac{\partial}{\partial a'_k} f_{\boldsymbol{x}k}^{a'_k-1}(1 - f_{\boldsymbol{x}k})^{b_k-1} \mathrm{d}f_{\boldsymbol{x}k}, \tag{22}$$

$$= -\sum_{k=1}^K \frac{1}{\mathrm{B}(a_k, b_k)} \frac{\partial}{\partial a'_k} \int_0^1 f_{\boldsymbol{x}k}^{a'_k-1}(1 - f_{\boldsymbol{x}k})^{b_k-1} \mathrm{d}f_{\boldsymbol{x}k}, \tag{23}$$

$$= -\sum_{k=1}^K \frac{1}{\mathrm{B}(a_k, b_k)} \frac{\partial}{\partial a'_k} \mathrm{B}(a'_k, b_k), \tag{24}$$

$$= -\sum_{k=1}^K \frac{1}{\mathrm{B}(a_k, b_k)} \mathrm{B}(a'_k, b_k)(\psi(a'_k) - \psi(a'_k + b_k)), \tag{25}$$

$$= -\sum_{k=1}^K \frac{1}{\mathrm{B}(a_k, b_k)} \frac{a_k}{a_k + b_k} \mathrm{B}(a_k, b_k)(\psi(a_k + 1) - \psi(a_k + 1 + b_k)), \tag{26}$$

$$= -\sum_{k=1}^K \frac{a_k}{a_k + b_k}(\psi(a_k + 1) - \psi(a_k + 1 + b_k)), \tag{27}$$

$$= -\sum_{k=1}^K \frac{\alpha_k}{z}\left(\psi\left(\alpha_k + 1\right) - \psi\left(z + 1\right)\right). \tag{28}$$

## F    IMPLEMENTATION DETAILS AND COMPUTATIONAL COMPLEXITY

We implemented all models using the `PyTorch` library (Paszke et al., 2019) and all experiments were conducted using a i5-6600K CPU, a GTX1070 GPU and a GTX2080 GPU with less than 300 hours of total runtime.

The Two Moons data was generated by the `make_moons` function from the `scikit-learn` library using 100 samples, 0.2 noise and random state 456, the `PyTorch` manual seed was set to 123. For this toy problem, all models were MLPs with two hidden layers consisting of 25 hidden units each, bias terms enabled and ReLU activation. For the Dropout models, the dropout rate was set to 0.2 and for the Ensemble models, we used 10 members per ensemble. All models were trained for 1000 epochs at a learning rate of 0.005 using the Adam optimizer (Kingma & Ba, 2015) with default parameters. The index set for the KL divergence was the visible 2D input plane, discretized at steps of 0.05. Since there was no mini-batch training, the KL term was not scaled according to Section B.1. This toy experiment is the only exception in this regard.

For Rotated MNIST, we used all 60000 images of shape 28x28x1 reshaped to 784 from the train split with pixel values normalized to $[-1, 1]$ and no other pre-processing or data augmentation. All 10000 images from the test set were used during evaluation, rotated by a fixed degree, ranging from 0° to 180° in 10° steps, resulting in a total of 190000 test images. The MNIST (LeCun et al., 2010) data is available under the terms of the Creative Commons Attribution-Share Alike 3.0 license. All models were MLPs with two hidden layers consisting of 50 units each, bias terms enabled and ReLU activation. For the Dropout models, the dropout rate was set to 0.2 and for the Ensemble models, we used 10 members per ensemble. All models were trained for 30 epochs at a learning rate of 0.001 using a mini-batch size of 256. The index set for the KL divergence was the training data itself, except for the index set comparison section, where the different index sets are stated explicitly. Results were obtained using 10 random seeds.

For corrupted CIFAR10 and CIFAR100, we used all 50000 images of shape 32x32x3 from the regular train splits. Following He et al. (2016), we normalized pixel values using the empirical mean and standard deviation, and employed data augmentation during training by first selecting random crops of size 32x32x3 after adding 4 pixels of zero padding to each side and then randomly flipping 50% of the images horizontally. All 10000 images from the regular test set were used during evaluation plus their corrupted versions (Hendrycks & Dietterich, 2019) with 19 different corruptions and 5 levels of severity, resulting in a total of 960000 test images. The CIFAR10 and CIFAR100 (Krizhevsky, 2009) data is available under the terms of the MIT License and the corrupted CIFAR10 and corrupted CIFAR100 (Hendrycks & Dietterich, 2019) data is available under the terms of the Apache License 2.0. All models were CNNs following the ResNet-18 architecture (He et al., 2016), designed for CIFAR images, rather than ImageNet.

Adopting He et al. (2016), we trained with a batch size of 128 and used the SGD optimizer with momentum (0.9) for 200 epochs and scaled the learning rate by 0.1 at epochs 100 and 150. For the MAP, MC Dropout and Ensemble models without fVI, we used 0.0005 weight decay. For MC Dropout models (Gal & Ghahramani, 2016), the dropout rate was set to 0.2. For Ensemble (Lakshminarayanan et al., 2017) and Ensemble fVI, we used 5 members per ensemble.

For Radial BNNs (Farquhar et al., 2020) and Radial fVI, we implemented weight priors for all convolutional weights but not for the final linear layer. The standard deviation $\sigma$ was parameterized using $\sigma = \log(1 + \exp(\rho))$ and $\rho$ was initialized to -5 while the means were initialized using the `PyTorch` default initialization scheme for CNNs. For Radial BNNs without fVI, we used a closed-form Gaussian weight KL divergence with a Gaussian prior with a mean of 0 and a standard deviation of 0.1. For Radial fVI, we used our fKL instead of the weight-space KL.

For Rank1 BNNs (Dusenberry et al., 2020) and Rank1 fVI, we used 4 ensemble members and 250 training epochs instead of 200 due to slow convergence and scaled the learning rate by 0.1 at epochs 150 and 200. During training, we used implicit batch ensembling (Wen et al., 2020), whereas during prediction, we created explicit ensemble predictions by replicating the input. We placed Rank1 Gaussian distributions over all convolutional weights but not over the final linear layer. The standard deviation $\sigma$ was parameterized using $\sigma = \log(1 + \exp(\rho))$ and $\rho$ was initialized to -3 while the means were initialized to 1. For Rank1 BNNs without fVI, following Dusenberry et al. (2020), the Rank1 priors were Gaussian with a mean of 1 and a standard deviation of 0.1, and weight decay of

0.0001 was used. We did not use KL annealing epochs. For Rank1 fVI, we used our fKL instead of the weight-space KL. For all fVI models, the index set for the KL divergence during fVI training was always the training data itself. Results were obtained using 10 random seeds.

To ensure numerical stability, we defined a minimum and maximum precision for the posterior Dirichlet estimation: $z_{\min} = K$ and $z_{\max} = N$, where $K$ is the number of classes and $N$ is the number of training examples. For the MAP models, we skipped the Dirichlet MLE and set $z = z_{\max}$ because $M = 1$. Similarly, we used $M = 1$ for the MC Dropout and Radial BNN models during training and also set $z = z_{\max}$, although we set $M = 10$ during evaluation. For the Ensemble models, $M$ was always the number of members in the ensemble and for the Rank1 models, we replicated the input M times during evaluation, which results in M distinct predictions. Furthermore, we applied a small amount of label smoothing $f_{\boldsymbol{x}\,k}^{(m)} \approx (1 - \gamma)f_{\boldsymbol{x}\,k}^{(m)} + \gamma\frac{1}{K}$ throughout all steps of the KL divergence estimation, where $\gamma$ was set to $10^{-4}$ for our experiments.

The equations from Minka's quasi-Newton maximum likelihood Dirichlet precision estimator (Minka, 2000), which we used for our implementation, translated to our notation are

$$\frac{1}{z} = \frac{1}{z} + \frac{1}{z^2}\frac{\Delta_1}{\Delta_2}, \qquad \bar{\alpha}_k = \frac{1}{M}\sum_{m=1}^{M} f_{\boldsymbol{x}\,k}^{(m)}, \qquad \breve{\alpha}_k = \frac{1}{M}\sum_{m=1}^{M} \log f_{\boldsymbol{x}\,k}^{(m)}, \qquad (29)$$

$$\Delta_1 = M\left(\psi_0(z) - \sum_{k=1}^{K}\bar{\alpha}_k\psi_0(z\bar{\alpha}_k) + \sum_{k=1}^{K}\bar{\alpha}_k\breve{\alpha}_k\right), \qquad (30)$$

$$\Delta_2 = M\left(\psi_1(z) - \sum_{k=1}^{K}\bar{\alpha}_k^2\psi_1(z\bar{\alpha}_k)\right), \qquad (31)$$

where $\psi_0$ is the digamma function and $\psi_1$ is the trigamma function.

We initialized the algorithm with an approximate maximum likelihood solution using Stirling's approximation to the gamma function $\Gamma$, which was also presented by Minka (2000),

$$z^{(0)} = \frac{K - 1}{-2\sum_{k=1}^{K}\bar{\alpha}_k(\log\breve{\alpha}_k - \log\bar{\alpha}_k)}. \qquad (32)$$

We stopped the algorithm once the change per step is less than $10^{-5}$. Counting the number of iterations until convergence for a trained Ensemble fVI model with $M = 10$ ensemble members and 10000 MNIST test examples, the mean was 3.0796, the 95th quantile was 3, the 99th quantile was 15 and the maximum was 1172. Note that the number of iterations until convergence in vectorized mini-batch computation is equal to the maximum number of iterations until convergence of the items in the mini-batch.

Although the computational complexity of the underlying deep learning model depends on the model architecture, data input size, number of parameters, etc., for the following comparison, we assume that a single forward pass through the model takes $\mathcal{O}(1)$, i.e. a constant amount of time, because the weight-space and function-space objectives share the same model. With a mini-batch size of $B$, computation of the standard ML objective takes $\mathcal{O}(BMK)$ time per mini-batch iteration. Assuming a constant number of quasi-Newton steps, the Dirichlet precision estimation takes $\mathcal{O}(SK + M)$ time for an index set of size $S$. Computing the fKL for an index set of size $S$ takes $\mathcal{O}(SMK)$ time. If the training data is used as index set the forward pass through the model can be shared between the log-likelihood and fKL calculation, resulting in an overall asymptotic time complexity of $\mathcal{O}(BMK)$ per mini-batch iteration. In case of a different index set, the asymptotic time complexity becomes $\mathcal{O}((B+S)MK)$.

# G EXPERIMENTAL RESULTS

All experimental results are reported here.

## G.1 ROTATED MNIST

Table 2: Accuracies for the rotated MNIST experiment. Means and standard errors over ten seeds. Best results within archetype in boldface, best results overall in blue.

| MNIST Accuracy ↑ | Angle ° | | | | | | |
|---|---|---|---|---|---|---|---|
| | 0 | 10 | 20 | 30 | 40 | 50 | 60 |
| MAP | $96.82 \pm 0.06$ | $94.19 \pm 0.15$ | $87.17 \pm 0.43$ | $71.60 \pm 0.76$ | $53.11 \pm 0.76$ | $\mathbf{36.03 \pm 0.52}$ | $\mathbf{25.18 \pm 0.46}$ |
| MAP fVI | $\mathbf{97.09 \pm 0.05}$ | $\mathbf{95.01 \pm 0.13}$ | $\mathbf{88.51 \pm 0.32}$ | $\mathbf{73.53 \pm 0.51}$ | $\mathbf{54.32 \pm 0.57}$ | $36.03 \pm 0.40$ | $24.56 \pm 0.35$ |
| MC Dropout | $\mathbf{96.23 \pm 0.05}$ | $93.63 \pm 0.16$ | $\mathbf{86.77 \pm 0.16}$ | $\mathbf{71.06 \pm 0.28}$ | $\mathbf{52.46 \pm 0.48}$ | $\mathbf{35.57 \pm 0.56}$ | $25.28 \pm 0.54$ |
| MC Dropout fVI | $96.10 \pm 0.08$ | $\mathbf{93.70 \pm 0.17}$ | $86.61 \pm 0.42$ | $71.04 \pm 0.74$ | $52.20 \pm 0.71$ | $34.94 \pm 0.55$ | $\mathbf{25.38 \pm 0.33}$ |
| Ensemble | $97.97 \pm 0.02$ | $96.36 \pm 0.04$ | $91.02 \pm 0.18$ | $76.98 \pm 0.25$ | $57.96 \pm 0.29$ | $39.52 \pm 0.27$ | $27.94 \pm 0.32$ |
| Ensemble fVI | $\mathbf{98.09 \pm 0.02}$ | $\mathbf{96.71 \pm 0.02}$ | $\mathbf{91.95 \pm 0.13}$ | $\mathbf{78.87 \pm 0.23}$ | $\mathbf{60.08 \pm 0.31}$ | $\mathbf{40.81 \pm 0.22}$ | $\mathbf{28.20 \pm 0.29}$ |

Table 3: Accuracies for the rotated MNIST experiment. Means and standard errors over ten seeds. Best results within archetype in boldface, best results overall in blue.

| MNIST Accuracy ↑ | Angle ° | | | | | |
|---|---|---|---|---|---|---|
| | 70 | 80 | 90 | 100 | 110 | 120 |
| MAP | $\mathbf{18.60 \pm 0.39}$ | $\mathbf{15.01 \pm 0.34}$ | $12.81 \pm 0.46$ | $11.77 \pm 0.52$ | $11.66 \pm 0.49$ | $14.09 \pm 0.33$ |
| MAP fVI | $17.91 \pm 0.42$ | $14.72 \pm 0.47$ | $\mathbf{13.33 \pm 0.48}$ | $\mathbf{13.02 \pm 0.40}$ | $\mathbf{13.33 \pm 0.28}$ | $\mathbf{15.38 \pm 0.21}$ |
| MC Dropout | $20.26 \pm 0.48$ | $17.22 \pm 0.47$ | $15.11 \pm 0.43$ | $13.41 \pm 0.38$ | $12.67 \pm 0.35$ | $14.32 \pm 0.23$ |
| MC Dropout fVI | $\mathbf{20.99 \pm 0.39}$ | $\mathbf{18.22 \pm 0.34}$ | $\mathbf{16.26 \pm 0.39}$ | $\mathbf{14.82 \pm 0.46}$ | $\mathbf{13.69 \pm 0.38}$ | $\mathbf{14.44 \pm 0.29}$ |
| Ensemble | $19.90 \pm 0.31$ | $16.11 \pm 0.32$ | $13.62 \pm 0.31$ | $12.09 \pm 0.26$ | $12.31 \pm 0.22$ | $14.92 \pm 0.21$ |
| Ensemble fVI | $\mathbf{20.08 \pm 0.38}$ | $\mathbf{16.43 \pm 0.37}$ | $\mathbf{14.28 \pm 0.30}$ | $\mathbf{13.07 \pm 0.21}$ | $\mathbf{13.19 \pm 0.16}$ | $\mathbf{15.43 \pm 0.12}$ |

Table 4: Accuracies for the rotated MNIST experiment. Means and standard errors over ten seeds. Best results within archetype in boldface, best results overall in blue.

| MNIST Accuracy ↑ | Angle ° | | | | | |
|---|---|---|---|---|---|---|
| | 130 | 140 | 150 | 160 | 170 | 180 |
| MAP | $16.92 \pm 0.43$ | $19.05 \pm 0.56$ | $22.36 \pm 0.55$ | $25.06 \pm 0.64$ | $26.95 \pm 0.58$ | $28.58 \pm 0.57$ |
| MAP fVI | $\mathbf{17.94 \pm 0.29}$ | $\mathbf{19.82 \pm 0.43}$ | $\mathbf{22.45 \pm 0.54}$ | $\mathbf{25.26 \pm 0.46}$ | $\mathbf{27.42 \pm 0.43}$ | $\mathbf{28.93 \pm 0.42}$ |
| MC Dropout | $\mathbf{17.35 \pm 0.34}$ | $20.11 \pm 0.49$ | $22.82 \pm 0.43$ | $24.37 \pm 0.39$ | $26.03 \pm 0.43$ | $28.14 \pm 0.43$ |
| MC Dropout fVI | $17.12 \pm 0.35$ | $\mathbf{20.25 \pm 0.40}$ | $\mathbf{23.25 \pm 0.34}$ | $\mathbf{25.24 \pm 0.27}$ | $\mathbf{26.86 \pm 0.18}$ | $\mathbf{28.95 \pm 0.29}$ |
| Ensemble | $18.02 \pm 0.32$ | $20.83 \pm 0.29$ | $24.49 \pm 0.26$ | $27.61 \pm 0.20$ | $30.03 \pm 0.24$ | $31.27 \pm 0.32$ |
| Ensemble fVI | $\mathbf{18.47 \pm 0.19}$ | $\mathbf{21.49 \pm 0.20}$ | $\mathbf{25.21 \pm 0.21}$ | $\mathbf{28.54 \pm 0.24}$ | $\mathbf{31.31 \pm 0.29}$ | $\mathbf{32.51 \pm 0.31}$ |

Table 5: Log-likelihoods for the rotated MNIST experiment. Means and standard errors over ten seeds. Best results within archetype in boldface, best results overall in blue.

| MNIST Log-Likelihood ↑ | Angle ° | | | | | | |
|---|---|---|---|---|---|---|---|
| | 0 | 10 | 20 | 30 | 40 | 50 | 60 |
| MAP | $\mathbf{-0.11 \pm 0.00}$ | $\mathbf{-0.20 \pm 0.00}$ | $-0.48 \pm 0.02$ | $-1.20 \pm 0.03$ | $-2.36 \pm 0.04$ | $-3.80 \pm 0.04$ | $-5.05 \pm 0.05$ |
| MAP fVI | $-0.13 \pm 0.00$ | $-0.20 \pm 0.00$ | $\mathbf{-0.39 \pm 0.01}$ | $\mathbf{-0.84 \pm 0.02}$ | $\mathbf{-1.53 \pm 0.02}$ | $\mathbf{-2.34 \pm 0.03}$ | $\mathbf{-3.02 \pm 0.03}$ |
| MC Dropout | $\mathbf{-0.15 \pm 0.00}$ | $\mathbf{-0.23 \pm 0.00}$ | $\mathbf{-0.43 \pm 0.00}$ | $\mathbf{-0.88 \pm 0.01}$ | $-1.50 \pm 0.02$ | $-2.28 \pm 0.03$ | $-2.97 \pm 0.04$ |
| MC Dropout fVI | $-0.20 \pm 0.00$ | $-0.28 \pm 0.01$ | $-0.48 \pm 0.01$ | $-0.91 \pm 0.02$ | $\mathbf{-1.46 \pm 0.02}$ | $\mathbf{-2.06 \pm 0.02}$ | $\mathbf{-2.50 \pm 0.02}$ |
| Ensemble | $\mathbf{-0.07 \pm 0.00}$ | $\mathbf{-0.12 \pm 0.00}$ | $\mathbf{-0.29 \pm 0.00}$ | $-0.73 \pm 0.01$ | $-1.50 \pm 0.02$ | $-2.56 \pm 0.03$ | $-3.60 \pm 0.04$ |
| Ensemble fVI | $-0.11 \pm 0.00$ | $-0.17 \pm 0.00$ | $-0.32 \pm 0.00$ | $\mathbf{-0.67 \pm 0.01}$ | $\mathbf{-1.20 \pm 0.01}$ | $\mathbf{-1.85 \pm 0.01}$ | $\mathbf{-2.43 \pm 0.01}$ |

Table 6: Log-likelihoods for the rotated MNIST experiment. Means and standard errors over ten seeds. Best results within archetype in boldface, best results overall in blue.

| MNIST Log-Likelihood ↑ | Angle ° 70 | 80 | 90 | 100 | 110 | 120 |
|---|---|---|---|---|---|---|
| MAP | $-6.05 \pm 0.06$ | $-6.79 \pm 0.07$ | $-7.22 \pm 0.09$ | $-7.34 \pm 0.09$ | $-7.27 \pm 0.10$ | $-6.95 \pm 0.09$ |
| MAP fVI | $\mathbf{-3.54 \pm 0.06}$ | $\mathbf{-3.94 \pm 0.07}$ | $\mathbf{-4.14 \pm 0.08}$ | $\mathbf{-4.18 \pm 0.08}$ | $\mathbf{-4.16 \pm 0.07}$ | $\mathbf{-4.05 \pm 0.06}$ |
| MC Dropout | $-3.47 \pm 0.04$ | $-3.88 \pm 0.04$ | $-4.17 \pm 0.05$ | $-4.28 \pm 0.05$ | $-4.32 \pm 0.05$ | $-4.15 \pm 0.05$ |
| MC Dropout fVI | $\color{blue}\mathbf{-2.81 \pm 0.02}$ | $\color{blue}\mathbf{-3.04 \pm 0.03}$ | $\color{blue}\mathbf{-3.20 \pm 0.04}$ | $\color{blue}\mathbf{-3.25 \pm 0.04}$ | $\color{blue}\mathbf{-3.28 \pm 0.04}$ | $\color{blue}\mathbf{-3.17 \pm 0.03}$ |
| Ensemble | $-4.52 \pm 0.04$ | $-5.31 \pm 0.04$ | $-5.77 \pm 0.05$ | $-5.90 \pm 0.06$ | $-5.86 \pm 0.06$ | $-5.68 \pm 0.06$ |
| Ensemble fVI | $\mathbf{-2.89 \pm 0.02}$ | $\mathbf{-3.27 \pm 0.02}$ | $\mathbf{-3.49 \pm 0.02}$ | $\mathbf{-3.57 \pm 0.02}$ | $\mathbf{-3.57 \pm 0.02}$ | $\mathbf{-3.47 \pm 0.02}$ |

Table 7: Log-likelihoods for the rotated MNIST experiment. Means and standard errors over ten seeds. Best results within archetype in boldface, best results overall in blue.

| MNIST Log-Likelihood ↑ | Angle ° 130 | 140 | 150 | 160 | 170 | 180 |
|---|---|---|---|---|---|---|
| MAP | $-6.57 \pm 0.09$ | $-6.36 \pm 0.08$ | $-6.20 \pm 0.07$ | $-6.16 \pm 0.06$ | $-6.13 \pm 0.05$ | $-6.00 \pm 0.07$ |
| MAP fVI | $\mathbf{-3.87 \pm 0.05}$ | $\mathbf{-3.78 \pm 0.04}$ | $\mathbf{-3.68 \pm 0.04}$ | $\mathbf{-3.63 \pm 0.03}$ | $\mathbf{-3.61 \pm 0.04}$ | $\mathbf{-3.50 \pm 0.04}$ |
| MC Dropout | $-3.93 \pm 0.05$ | $-3.85 \pm 0.06$ | $-3.87 \pm 0.05$ | $-4.04 \pm 0.04$ | $-4.15 \pm 0.04$ | $-4.19 \pm 0.04$ |
| MC Dropout fVI | $\color{blue}\mathbf{-3.03 \pm 0.02}$ | $\color{blue}\mathbf{-2.94 \pm 0.02}$ | $\color{blue}\mathbf{-2.91 \pm 0.02}$ | $\color{blue}\mathbf{-2.97 \pm 0.02}$ | $\color{blue}\mathbf{-3.00 \pm 0.01}$ | $\color{blue}\mathbf{-2.98 \pm 0.01}$ |
| Ensemble | $-5.37 \pm 0.05$ | $-5.24 \pm 0.04$ | $-5.18 \pm 0.02$ | $-5.31 \pm 0.01$ | $-5.37 \pm 0.02$ | $-5.25 \pm 0.02$ |
| Ensemble fVI | $\mathbf{-3.29 \pm 0.02}$ | $\mathbf{-3.22 \pm 0.01}$ | $\mathbf{-3.17 \pm 0.01}$ | $\mathbf{-3.21 \pm 0.02}$ | $\mathbf{-3.23 \pm 0.02}$ | $\mathbf{-3.16 \pm 0.02}$ |

Table 8: Expected calibration errors for the rotated MNIST experiment. Means and standard errors over ten seeds. Best results within archetype in boldface, best results overall in blue.

| MNIST ECE ↓ | Angle ° 0 | 10 | 20 | 30 | 40 | 50 | 60 |
|---|---|---|---|---|---|---|---|
| MAP | $\color{blue}\mathbf{0.01 \pm 0.00}$ | $\mathbf{0.02 \pm 0.00}$ | $0.06 \pm 0.00$ | $0.17 \pm 0.01$ | $0.32 \pm 0.01$ | $0.46 \pm 0.00$ | $0.56 \pm 0.00$ |
| MAP fVI | $0.05 \pm 0.00$ | $0.05 \pm 0.00$ | $\mathbf{0.04 \pm 0.00}$ | $\mathbf{0.03 \pm 0.00}$ | $\mathbf{0.15 \pm 0.01}$ | $\mathbf{0.28 \pm 0.01}$ | $\mathbf{0.38 \pm 0.01}$ |
| MC Dropout | $\mathbf{0.04 \pm 0.00}$ | $\mathbf{0.05 \pm 0.00}$ | $\mathbf{0.05 \pm 0.00}$ | $\color{blue}\mathbf{0.02 \pm 0.00}$ | $0.13 \pm 0.01$ | $0.25 \pm 0.01$ | $0.33 \pm 0.01$ |
| MC Dropout fVI | $0.09 \pm 0.00$ | $0.10 \pm 0.00$ | $0.10 \pm 0.00$ | $0.04 \pm 0.01$ | $\mathbf{0.07 \pm 0.01}$ | $\mathbf{0.19 \pm 0.01}$ | $\mathbf{0.27 \pm 0.01}$ |
| Ensemble | $\mathbf{0.01 \pm 0.00}$ | $\color{blue}\mathbf{0.02 \pm 0.00}$ | $\color{blue}\mathbf{0.01 \pm 0.00}$ | $\mathbf{0.04 \pm 0.00}$ | $0.15 \pm 0.00$ | $0.27 \pm 0.00$ | $0.36 \pm 0.00$ |
| Ensemble fVI | $0.06 \pm 0.00$ | $0.08 \pm 0.00$ | $0.10 \pm 0.00$ | $0.08 \pm 0.00$ | $\color{blue}\mathbf{0.02 \pm 0.00}$ | $\color{blue}\mathbf{0.14 \pm 0.00}$ | $\color{blue}\mathbf{0.23 \pm 0.00}$ |

Table 9: Expected calibration errors for the rotated MNIST experiment. Means and standard errors over ten seeds. Best results within archetype in boldface, best results overall in blue.

| MNIST ECE ↓ | Angle ° 70 | 80 | 90 | 100 | 110 | 120 |
|---|---|---|---|---|---|---|
| MAP | $0.62 \pm 0.00$ | $0.65 \pm 0.01$ | $0.68 \pm 0.01$ | $0.69 \pm 0.01$ | $0.69 \pm 0.01$ | $0.67 \pm 0.01$ |
| MAP fVI | $\mathbf{0.44 \pm 0.01}$ | $\mathbf{0.48 \pm 0.01}$ | $\mathbf{0.50 \pm 0.01}$ | $\mathbf{0.50 \pm 0.01}$ | $\mathbf{0.51 \pm 0.01}$ | $\mathbf{0.50 \pm 0.00}$ |
| MC Dropout | $0.39 \pm 0.01$ | $0.43 \pm 0.01$ | $0.46 \pm 0.01$ | $0.48 \pm 0.01$ | $0.50 \pm 0.01$ | $0.48 \pm 0.01$ |
| MC Dropout fVI | $\mathbf{0.32 \pm 0.01}$ | $\mathbf{0.36 \pm 0.01}$ | $\mathbf{0.38 \pm 0.01}$ | $\color{blue}\mathbf{0.40 \pm 0.01}$ | $\mathbf{0.42 \pm 0.01}$ | $\mathbf{0.40 \pm 0.01}$ |
| Ensemble | $0.43 \pm 0.00$ | $0.46 \pm 0.00$ | $0.49 \pm 0.00$ | $0.51 \pm 0.01$ | $0.52 \pm 0.00$ | $0.50 \pm 0.00$ |
| Ensemble fVI | $\color{blue}\mathbf{0.31 \pm 0.00}$ | $\color{blue}\mathbf{0.35 \pm 0.00}$ | $\color{blue}\mathbf{0.38 \pm 0.00}$ | $\mathbf{0.40 \pm 0.00}$ | $\color{blue}\mathbf{0.41 \pm 0.00}$ | $\color{blue}\mathbf{0.40 \pm 0.00}$ |

Table 10: Expected calibration errors for the rotated MNIST experiment. Means and standard errors over ten seeds. Best results within archetype in boldface, best results overall in blue.

| MNIST ECE ↓ | Angle ° 130 | 140 | 150 | 160 | 170 | 180 |
|---|---|---|---|---|---|---|
| MAP | $0.65 \pm 0.01$ | $0.63 \pm 0.01$ | $0.60 \pm 0.01$ | $0.58 \pm 0.01$ | $0.57 \pm 0.01$ | $0.56 \pm 0.01$ |
| MAP fVI | $\mathbf{0.48 \pm 0.00}$ | $\mathbf{0.47 \pm 0.01}$ | $\mathbf{0.45 \pm 0.01}$ | $\mathbf{0.43 \pm 0.01}$ | $\mathbf{0.41 \pm 0.01}$ | $\mathbf{0.40 \pm 0.01}$ |
| MC Dropout | $0.45 \pm 0.01$ | $0.42 \pm 0.01$ | $0.41 \pm 0.01$ | $0.41 \pm 0.01$ | $0.41 \pm 0.01$ | $0.40 \pm 0.00$ |
| MC Dropout fVI | $\color{blue}\mathbf{0.37 \pm 0.00}$ | $\mathbf{0.35 \pm 0.00}$ | $\mathbf{0.33 \pm 0.00}$ | $\mathbf{0.34 \pm 0.00}$ | $\mathbf{0.33 \pm 0.00}$ | $\mathbf{0.32 \pm 0.00}$ |
| Ensemble | $0.48 \pm 0.00$ | $0.46 \pm 0.00$ | $0.44 \pm 0.00$ | $0.43 \pm 0.00$ | $0.42 \pm 0.00$ | $0.42 \pm 0.00$ |
| Ensemble fVI | $\mathbf{0.38 \pm 0.00}$ | $\mathbf{0.35 \pm 0.00}$ | $\color{blue}\mathbf{0.33 \pm 0.00}$ | $\color{blue}\mathbf{0.32 \pm 0.00}$ | $\color{blue}\mathbf{0.31 \pm 0.00}$ | $\color{blue}\mathbf{0.30 \pm 0.00}$ |

## G.2  CIFAR10

Table 11: Accuracies for the corrupted CIFAR10 experiment. Means and standard errors over ten seeds. Best results within archetype in boldface, best results overall in blue.

| CIFAR10 Accuracy ↑ | Corruption Severity | | | | | |
|---|---|---|---|---|---|---|
| | 0 | 1 | 2 | 3 | 4 | 5 |
| MAP | $94.32 \pm 0.05$ | $87.65 \pm 0.08$ | $\mathbf{81.76 \pm 0.09}$ | $\mathbf{75.97 \pm 0.13}$ | $\mathbf{68.86 \pm 0.20}$ | $\mathbf{57.28 \pm 0.21}$ |
| MAP fVI | $\mathbf{94.40 \pm 0.08}$ | $\mathbf{87.66 \pm 0.06}$ | $81.66 \pm 0.10$ | $75.81 \pm 0.16$ | $68.77 \pm 0.17$ | $57.06 \pm 0.18$ |
| MC Dropout | $\mathbf{94.32 \pm 0.04}$ | $\mathbf{88.21 \pm 0.07}$ | $\mathbf{82.13 \pm 0.13}$ | $\mathbf{75.81 \pm 0.19}$ | $\mathbf{67.59 \pm 0.21}$ | $\mathbf{55.72 \pm 0.26}$ |
| MC Dropout fVI | $93.38 \pm 0.03$ | $87.01 \pm 0.09$ | $80.64 \pm 0.14$ | $74.36 \pm 0.18$ | $66.33 \pm 0.18$ | $54.87 \pm 0.19$ |
| Ensemble | $\color{blue}{\mathbf{95.30 \pm 0.04}}$ | $89.37 \pm 0.03$ | $83.77 \pm 0.06$ | $78.21 \pm 0.07$ | $71.05 \pm 0.11$ | $59.33 \pm 0.16$ |
| Ensemble fVI | $95.26 \pm 0.03$ | $\color{blue}{\mathbf{89.44 \pm 0.03}}$ | $\color{blue}{\mathbf{83.94 \pm 0.07}}$ | $\color{blue}{\mathbf{78.49 \pm 0.08}}$ | $\color{blue}{\mathbf{71.42 \pm 0.12}}$ | $\color{blue}{\mathbf{59.73 \pm 0.16}}$ |
| Radial | $\mathbf{95.05 \pm 0.04}$ | $\mathbf{87.98 \pm 0.07}$ | $\mathbf{81.82 \pm 0.10}$ | $75.93 \pm 0.12$ | $\mathbf{68.93 \pm 0.16}$ | $57.24 \pm 0.21$ |
| Radial fVI | $93.73 \pm 0.03$ | $87.43 \pm 0.07$ | $81.75 \pm 0.14$ | $\mathbf{76.09 \pm 0.21}$ | $68.84 \pm 0.29$ | $\mathbf{57.42 \pm 0.35}$ |
| Rank1 | $93.68 \pm 0.05$ | $87.60 \pm 0.05$ | $82.32 \pm 0.08$ | $\mathbf{76.90 \pm 0.08}$ | $\mathbf{69.92 \pm 0.13}$ | $\mathbf{58.53 \pm 0.17}$ |
| Rank1 fVI | $\mathbf{93.91 \pm 0.04}$ | $\mathbf{87.75 \pm 0.05}$ | $\mathbf{82.38 \pm 0.09}$ | $76.77 \pm 0.14$ | $69.69 \pm 0.17$ | $58.23 \pm 0.19$ |
| Subnetwork | $91.00 \pm 0.00$ | $83.00 \pm 1.00$ | $77.00 \pm 0.00$ | $68.00 \pm 1.00$ | $64.00 \pm 1.00$ | $59.00 \pm 0.00$ |
| Belief Matching | $94.52 \pm 0.03$ | $86.98 \pm 0.12$ | $80.39 \pm 0.21$ | $73.62 \pm 0.29$ | $65.74 \pm 0.35$ | $53.67 \pm 0.36$ |
| Prior Networks | $66.65 \pm 0.61$ | $61.28 \pm 0.43$ | $57.87 \pm 0.37$ | $54.71 \pm 0.35$ | $50.24 \pm 0.33$ | $42.29 \pm 0.31$ |

Table 12: Log-likelihoods for the corrupted CIFAR10 experiment. Means and standard errors over ten seeds. Best results within archetype in boldface, best results overall in blue.

| CIFAR10 Log-Likelihood ↑ | Corruption Severity | | | | | |
|---|---|---|---|---|---|---|
| | 0 | 1 | 2 | 3 | 4 | 5 |
| MAP | $\mathbf{-0.22 \pm 0.00}$ | $-0.52 \pm 0.00$ | $-0.80 \pm 0.01$ | $-1.12 \pm 0.01$ | $-1.52 \pm 0.01$ | $-2.20 \pm 0.02$ |
| MAP fVI | $-0.25 \pm 0.00$ | $\mathbf{-0.48 \pm 0.00}$ | $\mathbf{-0.69 \pm 0.00}$ | $\mathbf{-0.90 \pm 0.01}$ | $\mathbf{-1.16 \pm 0.01}$ | $\mathbf{-1.60 \pm 0.01}$ |
| MC Dropout | $\mathbf{-0.17 \pm 0.00}$ | $\mathbf{-0.39 \pm 0.00}$ | $\mathbf{-0.63 \pm 0.01}$ | $-0.93 \pm 0.01$ | $-1.36 \pm 0.02$ | $-2.09 \pm 0.02$ |
| MC Dropout fVI | $-0.25 \pm 0.00$ | $-0.44 \pm 0.00$ | $-0.64 \pm 0.01$ | $\mathbf{-0.85 \pm 0.01}$ | $\mathbf{-1.12 \pm 0.01}$ | $\mathbf{-1.56 \pm 0.01}$ |
| Ensemble | $\color{blue}{\mathbf{-0.15 \pm 0.00}}$ | $\color{blue}{\mathbf{-0.35 \pm 0.00}}$ | $\color{blue}{\mathbf{-0.54 \pm 0.00}}$ | $-0.76 \pm 0.00$ | $-1.03 \pm 0.01$ | $-1.51 \pm 0.01$ |
| Ensemble fVI | $-0.21 \pm 0.00$ | $-0.38 \pm 0.00$ | $-0.55 \pm 0.00$ | $\color{blue}{\mathbf{-0.72 \pm 0.00}}$ | $\color{blue}{\mathbf{-0.94 \pm 0.00}}$ | $\color{blue}{\mathbf{-1.33 \pm 0.01}}$ |
| Radial | $\mathbf{-0.21 \pm 0.00}$ | $-0.58 \pm 0.00$ | $-0.93 \pm 0.01$ | $-1.32 \pm 0.01$ | $-1.79 \pm 0.01$ | $-2.61 \pm 0.02$ |
| Radial fVI | $-0.28 \pm 0.00$ | $\mathbf{-0.49 \pm 0.00}$ | $\mathbf{-0.69 \pm 0.01}$ | $\mathbf{-0.90 \pm 0.01}$ | $\mathbf{-1.17 \pm 0.01}$ | $\mathbf{-1.61 \pm 0.02}$ |
| Rank1 | $-0.33 \pm 0.00$ | $-0.71 \pm 0.01$ | $-1.06 \pm 0.01$ | $-1.46 \pm 0.01$ | $-2.02 \pm 0.02$ | $-3.00 \pm 0.03$ |
| Rank1 fVI | $\mathbf{-0.27 \pm 0.00}$ | $\mathbf{-0.47 \pm 0.00}$ | $\mathbf{-0.65 \pm 0.00}$ | $\mathbf{-0.85 \pm 0.01}$ | $\mathbf{-1.10 \pm 0.01}$ | $\mathbf{-1.53 \pm 0.01}$ |
| Subnetwork | $-0.27 \pm 0.00$ | $-0.51 \pm 0.01$ | $-0.73 \pm 0.01$ | $-1.06 \pm 0.02$ | $-1.25 \pm 0.03$ | $-1.47 \pm 0.03$ |
| Belief Matching | $-0.26 \pm 0.00$ | $-0.51 \pm 0.00$ | $-0.73 \pm 0.01$ | $-0.97 \pm 0.01$ | $-1.26 \pm 0.01$ | $-1.70 \pm 0.02$ |
| Prior Networks | $-1.45 \pm 0.02$ | $-1.59 \pm 0.01$ | $-1.66 \pm 0.01$ | $-1.73 \pm 0.01$ | $-1.85 \pm 0.01$ | $-2.06 \pm 0.01$ |

Table 13: Expected calibration errors for the corrupted CIFAR10 experiment. Means and standard errors over ten seeds. Best results within archetype in boldface, best results overall in blue.

| CIFAR10 ECE ↓ | Corruption Severity | | | | | |
|---|---|---|---|---|---|---|
| | 0 | 1 | 2 | 3 | 4 | 5 |
| MAP | $\mathbf{0.03 \pm 0.00}$ | $0.08 \pm 0.00$ | $0.12 \pm 0.00$ | $0.16 \pm 0.00$ | $0.22 \pm 0.00$ | $0.30 \pm 0.00$ |
| MAP fVI | $0.05 \pm 0.00$ | $\mathbf{0.05 \pm 0.00}$ | $\mathbf{0.05 \pm 0.00}$ | $\mathbf{0.09 \pm 0.00}$ | $\mathbf{0.14 \pm 0.00}$ | $\mathbf{0.22 \pm 0.00}$ |
| MC Dropout | $\mathbf{0.01 \pm 0.00}$ | $\mathbf{0.03 \pm 0.00}$ | $0.06 \pm 0.00$ | $0.09 \pm 0.00$ | $0.15 \pm 0.00$ | $0.23 \pm 0.00$ |
| MC Dropout fVI | $0.06 \pm 0.00$ | $0.03 \pm 0.00$ | $\mathbf{0.03 \pm 0.00}$ | $\mathbf{0.05 \pm 0.00}$ | $\mathbf{0.09 \pm 0.00}$ | $\mathbf{0.17 \pm 0.00}$ |
| Ensemble | $\color{blue}{\mathbf{0.01 \pm 0.00}}$ | $\color{blue}{\mathbf{0.02 \pm 0.00}}$ | $0.05 \pm 0.00$ | $0.08 \pm 0.00$ | $0.12 \pm 0.00$ | $0.19 \pm 0.00$ |
| Ensemble fVI | $0.07 \pm 0.00$ | $0.05 \pm 0.00$ | $\color{blue}{\mathbf{0.03 \pm 0.00}}$ | $\color{blue}{\mathbf{0.04 \pm 0.00}}$ | $\color{blue}{\mathbf{0.06 \pm 0.00}}$ | $\color{blue}{\mathbf{0.11 \pm 0.00}}$ |
| Radial | $\mathbf{0.03 \pm 0.00}$ | $0.08 \pm 0.00$ | $0.13 \pm 0.00$ | $0.17 \pm 0.00$ | $0.23 \pm 0.00$ | $0.32 \pm 0.00$ |
| Radial fVI | $0.05 \pm 0.00$ | $\mathbf{0.05 \pm 0.00}$ | $\mathbf{0.05 \pm 0.00}$ | $\mathbf{0.09 \pm 0.00}$ | $\mathbf{0.14 \pm 0.00}$ | $\mathbf{0.23 \pm 0.00}$ |
| Rank1 | $\mathbf{0.04 \pm 0.00}$ | $0.09 \pm 0.00$ | $0.13 \pm 0.00$ | $0.17 \pm 0.00$ | $0.23 \pm 0.00$ | $0.32 \pm 0.00$ |
| Rank1 fVI | $0.05 \pm 0.00$ | $\mathbf{0.05 \pm 0.00}$ | $\mathbf{0.04 \pm 0.00}$ | $\mathbf{0.07 \pm 0.00}$ | $\mathbf{0.12 \pm 0.00}$ | $\mathbf{0.20 \pm 0.00}$ |
| Subnetwork | $0.01 \pm 0.00$ | $0.03 \pm 0.00$ | $0.06 \pm 0.00$ | $0.11 \pm 0.01$ | $0.13 \pm 0.01$ | $0.16 \pm 0.01$ |
| Belief Matching | $0.07 \pm 0.00$ | $0.07 \pm 0.00$ | $0.07 \pm 0.00$ | $0.09 \pm 0.00$ | $0.14 \pm 0.00$ | $0.24 \pm 0.00$ |
| Prior Networks | $0.20 \pm 0.00$ | $0.21 \pm 0.00$ | $0.21 \pm 0.00$ | $0.22 \pm 0.00$ | $0.23 \pm 0.00$ | $0.24 \pm 0.00$ |

Table 14: Accuracies for the CIFAR10 adversarial attack experiment. Means and standard errors over ten seeds. Best results within archetype in boldface, best results overall in blue.

| CIFAR10 | Adversarial Attack Epsilon | | | | | | |
|---|---|---|---|---|---|---|---|
| Accuracy ↑ | 0.00 | 0.05 | 0.10 | 0.15 | 0.20 | 0.25 | 0.30 |
| MAP | $94.32 \pm 0.05$ | $42.35 \pm 0.23$ | $35.61 \pm 0.31$ | $31.93 \pm 0.38$ | $28.04 \pm 0.41$ | $23.60 \pm 0.40$ | $19.65 \pm 0.34$ |
| MAP fVI | $\mathbf{94.40 \pm 0.08}$ | $\mathbf{70.04 \pm 0.17}$ | $\mathbf{61.10 \pm 0.20}$ | $\mathbf{51.32 \pm 0.47}$ | $\mathbf{40.86 \pm 0.68}$ | $\mathbf{31.60 \pm 0.77}$ | $\mathbf{24.80 \pm 0.63}$ |
| MC Dropout | $\mathbf{94.32 \pm 0.02}$ | $43.30 \pm 0.15$ | $32.76 \pm 0.24$ | $29.39 \pm 0.25$ | $27.08 \pm 0.31$ | $23.83 \pm 0.38$ | $19.99 \pm 0.44$ |
| MC Dropout fVI | $93.43 \pm 0.04$ | $\mathbf{56.99 \pm 0.16}$ | $\mathbf{49.91 \pm 0.29}$ | $\mathbf{43.73 \pm 0.39}$ | $\mathbf{36.97 \pm 0.57}$ | $\mathbf{29.86 \pm 0.68}$ | $\mathbf{23.63 \pm 0.68}$ |
| Ensemble | $\mathbf{95.30 \pm 0.04}$ | $43.99 \pm 0.10$ | $29.06 \pm 0.13$ | $22.55 \pm 0.13$ | $18.58 \pm 0.15$ | $15.68 \pm 0.15$ | $13.54 \pm 0.15$ |
| Ensemble fVI | $95.26 \pm 0.03$ | $\mathbf{60.12 \pm 0.10}$ | $\mathbf{49.34 \pm 0.21}$ | $\mathbf{39.38 \pm 0.36}$ | $\mathbf{30.59 \pm 0.42}$ | $\mathbf{23.73 \pm 0.43}$ | $\mathbf{18.92 \pm 0.39}$ |
| Radial | $\mathbf{94.84 \pm 0.04}$ | $27.92 \pm 0.17$ | $18.50 \pm 0.18$ | $14.98 \pm 0.22$ | $12.69 \pm 0.25$ | $11.28 \pm 0.22$ | $10.47 \pm 0.20$ |
| Radial fVI | $93.72 \pm 0.03$ | $\mathbf{67.33 \pm 0.13}$ | $\mathbf{59.92 \pm 0.25}$ | $\mathbf{52.07 \pm 0.40}$ | $\mathbf{43.46 \pm 0.60}$ | $\mathbf{35.32 \pm 0.64}$ | $\mathbf{28.44 \pm 0.66}$ |
| Rank1 | $93.55 \pm 0.05$ | $19.65 \pm 0.21$ | $9.66 \pm 0.18$ | $8.34 \pm 0.16$ | $8.47 \pm 0.17$ | $8.77 \pm 0.20$ | $9.11 \pm 0.19$ |
| Rank1 fVI | $\mathbf{93.86 \pm 0.04}$ | $\mathbf{67.89 \pm 0.14}$ | $\mathbf{58.99 \pm 0.12}$ | $\mathbf{49.88 \pm 0.24}$ | $\mathbf{40.48 \pm 0.36}$ | $\mathbf{31.82 \pm 0.51}$ | $\mathbf{24.89 \pm 0.57}$ |

Table 15: Log-likelihoods for the CIFAR10 adversarial attack experiment. Means and standard errors over ten seeds. Best results within archetype in boldface, best results overall in blue.

| CIFAR10 | Adversarial Attack Epsilon | | | | | | |
|---|---|---|---|---|---|---|---|
| Log-Likelihood ↑ | 0.00 | 0.05 | 0.10 | 0.15 | 0.20 | 0.25 | 0.30 |
| MAP | $\mathbf{-0.22 \pm 0.00}$ | $-4.09 \pm 0.01$ | $-4.58 \pm 0.02$ | $-4.71 \pm 0.03$ | $-4.87 \pm 0.04$ | $-5.16 \pm 0.05$ | $-5.53 \pm 0.06$ |
| MAP fVI | $-0.25 \pm 0.00$ | $\mathbf{-1.36 \pm 0.01}$ | $\mathbf{-1.66 \pm 0.01}$ | $\mathbf{-1.97 \pm 0.02}$ | $\mathbf{-2.34 \pm 0.03}$ | $\mathbf{-2.71 \pm 0.04}$ | $\mathbf{-3.01 \pm 0.04}$ |
| MC Dropout | $\mathbf{-0.17 \pm 0.00}$ | $-3.26 \pm 0.01$ | $-4.42 \pm 0.02$ | $-4.61 \pm 0.02$ | $-4.60 \pm 0.02$ | $-4.69 \pm 0.05$ | $-4.96 \pm 0.07$ |
| MC Dropout fVI | $-0.25 \pm 0.00$ | $\mathbf{-1.82 \pm 0.01}$ | $\mathbf{-2.19 \pm 0.01}$ | $\mathbf{-2.37 \pm 0.02}$ | $\mathbf{-2.52 \pm 0.02}$ | $\mathbf{-2.70 \pm 0.03}$ | $\mathbf{-2.90 \pm 0.04}$ |
| Ensemble | $\mathbf{-0.15 \pm 0.00}$ | $-2.58 \pm 0.01$ | $-3.61 \pm 0.01$ | $-3.91 \pm 0.01$ | $-4.07 \pm 0.01$ | $-4.21 \pm 0.02$ | $-4.32 \pm 0.03$ |
| Ensemble fVI | $-0.21 \pm 0.00$ | $\mathbf{-1.49 \pm 0.00}$ | $\mathbf{-1.98 \pm 0.01}$ | $\mathbf{-2.30 \pm 0.01}$ | $\mathbf{-2.59 \pm 0.01}$ | $\mathbf{-2.82 \pm 0.02}$ | $\mathbf{-3.00 \pm 0.02}$ |
| Radial | $\mathbf{-0.21 \pm 0.00}$ | $-5.52 \pm 0.01$ | $-6.49 \pm 0.02$ | $-6.78 \pm 0.04$ | $-7.01 \pm 0.05$ | $-7.30 \pm 0.07$ | $-7.56 \pm 0.09$ |
| Radial fVI | $-0.28 \pm 0.00$ | $\mathbf{-1.50 \pm 0.01}$ | $\mathbf{-1.76 \pm 0.01}$ | $\mathbf{-2.01 \pm 0.01}$ | $\mathbf{-2.29 \pm 0.02}$ | $\mathbf{-2.57 \pm 0.03}$ | $\mathbf{-2.83 \pm 0.03}$ |
| Rank1 | $-0.33 \pm 0.00$ | $-7.84 \pm 0.02$ | $-9.30 \pm 0.03$ | $-9.39 \pm 0.03$ | $-9.23 \pm 0.04$ | $-9.08 \pm 0.06$ | $-9.00 \pm 0.07$ |
| Rank1 fVI | $\mathbf{-0.27 \pm 0.00}$ | $\mathbf{-1.46 \pm 0.01}$ | $\mathbf{-1.75 \pm 0.00}$ | $\mathbf{-2.04 \pm 0.01}$ | $\mathbf{-2.34 \pm 0.02}$ | $\mathbf{-2.66 \pm 0.03}$ | $\mathbf{-2.93 \pm 0.04}$ |

Table 16: Expected calibration errors for the CIFAR10 adversarial attack experiment. Means and standard errors over ten seeds. Best results within archetype in boldface, best results overall in blue.

| CIFAR10 | Adversarial Attack Epsilon | | | | | | |
|---|---|---|---|---|---|---|---|
| ECE ↓ | 0.00 | 0.05 | 0.10 | 0.15 | 0.20 | 0.25 | 0.30 |
| MAP | $\mathbf{0.03 \pm 0.00}$ | $0.48 \pm 0.00$ | $0.53 \pm 0.00$ | $0.55 \pm 0.00$ | $0.58 \pm 0.00$ | $0.61 \pm 0.00$ | $0.65 \pm 0.01$ |
| MAP fVI | $0.05 \pm 0.00$ | $\mathbf{0.19 \pm 0.00}$ | $\mathbf{0.24 \pm 0.00}$ | $\mathbf{0.30 \pm 0.00}$ | $\mathbf{0.38 \pm 0.01}$ | $\mathbf{0.45 \pm 0.01}$ | $\mathbf{0.52 \pm 0.01}$ |
| MC Dropout | $\mathbf{0.01 \pm 0.00}$ | $0.43 \pm 0.00$ | $0.53 \pm 0.00$ | $0.53 \pm 0.00$ | $0.52 \pm 0.00$ | $0.52 \pm 0.00$ | $0.54 \pm 0.01$ |
| MC Dropout fVI | $0.06 \pm 0.00$ | $\mathbf{0.28 \pm 0.00}$ | $\mathbf{0.32 \pm 0.00}$ | $\mathbf{0.34 \pm 0.00}$ | $\mathbf{0.35 \pm 0.00}$ | $\mathbf{0.38 \pm 0.01}$ | $\mathbf{0.41 \pm 0.01}$ |
| Ensemble | $\mathbf{0.01 \pm 0.00}$ | $0.39 \pm 0.00$ | $0.51 \pm 0.00$ | $0.54 \pm 0.00$ | $0.56 \pm 0.00$ | $0.57 \pm 0.00$ | $0.58 \pm 0.00$ |
| Ensemble fVI | $0.07 \pm 0.00$ | $\mathbf{0.20 \pm 0.00}$ | $\mathbf{0.27 \pm 0.00}$ | $\mathbf{0.32 \pm 0.00}$ | $\mathbf{0.37 \pm 0.00}$ | $\mathbf{0.41 \pm 0.00}$ | $\mathbf{0.45 \pm 0.00}$ |
| Radial | $\mathbf{0.03 \pm 0.00}$ | $0.62 \pm 0.00$ | $0.70 \pm 0.00$ | $0.71 \pm 0.00$ | $0.73 \pm 0.00$ | $0.74 \pm 0.00$ | $0.75 \pm 0.01$ |
| Radial fVI | $0.05 \pm 0.00$ | $\mathbf{0.21 \pm 0.00}$ | $\mathbf{0.26 \pm 0.00}$ | $\mathbf{0.30 \pm 0.00}$ | $\mathbf{0.36 \pm 0.00}$ | $\mathbf{0.42 \pm 0.01}$ | $\mathbf{0.47 \pm 0.01}$ |
| Rank1 | $\mathbf{0.04 \pm 0.00}$ | $0.76 \pm 0.00$ | $0.86 \pm 0.00$ | $0.86 \pm 0.00$ | $0.83 \pm 0.00$ | $0.81 \pm 0.00$ | $0.80 \pm 0.00$ |
| Rank1 fVI | $0.05 \pm 0.00$ | $\mathbf{0.20 \pm 0.00}$ | $\mathbf{0.25 \pm 0.00}$ | $\mathbf{0.30 \pm 0.00}$ | $\mathbf{0.36 \pm 0.00}$ | $\mathbf{0.43 \pm 0.01}$ | $\mathbf{0.49 \pm 0.01}$ |

## G.3 CIFAR100

Table 17: Accuracies for the corrupted CIFAR100 experiment. Means and standard errors over ten seeds. Best results within archetype in boldface, best results overall in blue.

| CIFAR100 Accuracy ↑ | Corruption Severity | | | | | |
|---|---|---|---|---|---|---|
| | 0 | 1 | 2 | 3 | 4 | 5 |
| MAP | **75.68 ± 0.07** | 64.17 ± 0.06 | 55.41 ± 0.07 | 49.78 ± 0.07 | 43.11 ± 0.08 | 33.03 ± 0.08 |
| MAP fVI | 74.77 ± 0.09 | **64.26 ± 0.07** | **55.82 ± 0.09** | **50.31 ± 0.11** | **43.56 ± 0.12** | **33.81 ± 0.11** |
| MC Dropout | **74.15 ± 0.07** | **63.23 ± 0.06** | **54.04 ± 0.09** | **48.33 ± 0.08** | **41.63 ± 0.08** | **32.02 ± 0.09** |
| MC Dropout fVI | 71.53 ± 0.12 | 61.03 ± 0.10 | 51.94 ± 0.11 | 46.46 ± 0.10 | 39.88 ± 0.09 | 30.87 ± 0.10 |
| Ensemble | **79.13 ± 0.05** | **68.00 ± 0.05** | **59.19 ± 0.06** | **53.42 ± 0.07** | **46.45 ± 0.06** | **35.69 ± 0.06** |
| Ensemble fVI | 75.89 ± 0.06 | 66.38 ± 0.07 | 57.97 ± 0.09 | 52.23 ± 0.09 | 45.14 ± 0.09 | 35.19 ± 0.11 |
| Radial | **76.40 ± 0.08** | 63.76 ± 0.07 | 54.68 ± 0.06 | 49.02 ± 0.05 | 42.29 ± 0.07 | 31.89 ± 0.07 |
| Radial fVI | 75.29 ± 0.10 | **64.84 ± 0.11** | **56.49 ± 0.12** | **50.96 ± 0.11** | **44.22 ± 0.10** | **34.43 ± 0.09** |
| Rank1 | 73.68 ± 0.10 | 63.48 ± 0.09 | 55.34 ± 0.12 | 49.92 ± 0.11 | 43.45 ± 0.10 | 33.87 ± 0.11 |
| Rank1 fVI | **75.56 ± 0.10** | **65.49 ± 0.08** | **57.55 ± 0.11** | **52.24 ± 0.10** | **45.63 ± 0.11** | **35.73 ± 0.11** |

Table 18: Log-likelihoods for the corrupted CIFAR100 experiment. Means and standard errors over ten seeds. Best results within archetype in boldface, best results overall in blue.

| CIFAR100 Log-Likelihood ↑ | Corruption Severity | | | | | |
|---|---|---|---|---|---|---|
| | 0 | 1 | 2 | 3 | 4 | 5 |
| MAP | **−1.00 ± 0.00** | **−1.59 ± 0.00** | −2.09 ± 0.01 | −2.48 ± 0.01 | −2.99 ± 0.01 | −3.73 ± 0.01 |
| MAP fVI | −1.20 ± 0.00 | −1.69 ± 0.00 | **−2.08 ± 0.01** | **−2.35 ± 0.01** | **−2.69 ± 0.01** | **−3.19 ± 0.01** |
| MC Dropout | **−0.97 ± 0.00** | **−1.51 ± 0.00** | **−2.02 ± 0.01** | −2.42 ± 0.01 | −2.96 ± 0.01 | −3.76 ± 0.02 |
| MC Dropout fVI | −1.17 ± 0.00 | −1.65 ± 0.01 | −2.09 ± 0.01 | **−2.39 ± 0.01** | **−2.79 ± 0.01** | **−3.35 ± 0.01** |
| Ensemble | **−0.81 ± 0.00** | **−1.30 ± 0.00** | **−1.70 ± 0.00** | **−1.98 ± 0.00** | **−2.37 ± 0.01** | **−2.93 ± 0.01** |
| Ensemble fVI | −1.18 ± 0.00 | −1.61 ± 0.00 | −1.98 ± 0.00 | −2.24 ± 0.00 | −2.58 ± 0.00 | −3.06 ± 0.01 |
| Radial | **−0.98 ± 0.00** | −1.66 ± 0.01 | −2.21 ± 0.01 | −2.65 ± 0.02 | −3.21 ± 0.02 | −4.09 ± 0.03 |
| Radial fVI | −1.21 ± 0.00 | **−1.69 ± 0.00** | **−2.08 ± 0.01** | **−2.34 ± 0.01** | **−2.67 ± 0.01** | **−3.16 ± 0.00** |
| Rank1 | −1.48 ± 0.01 | −2.29 ± 0.01 | −3.01 ± 0.01 | −3.59 ± 0.02 | −4.38 ± 0.02 | −5.57 ± 0.02 |
| Rank1 fVI | **−1.17 ± 0.00** | **−1.64 ± 0.00** | **−2.00 ± 0.01** | **−2.25 ± 0.01** | **−2.58 ± 0.01** | **−3.06 ± 0.01** |

Table 19: Expected calibration errors for the corrupted CIFAR100 experiment. Means and standard errors over ten seeds. Best results within archetype in boldface, best results overall in blue.

| CIFAR100 ECE ↓ | Corruption Severity | | | | | |
|---|---|---|---|---|---|---|
| | 0 | 1 | 2 | 3 | 4 | 5 |
| MAP | **0.08 ± 0.00** | 0.12 ± 0.00 | 0.16 ± 0.00 | 0.20 ± 0.00 | 0.23 ± 0.00 | 0.30 ± 0.00 |
| MAP fVI | 0.12 ± 0.00 | **0.11 ± 0.00** | **0.09 ± 0.00** | **0.06 ± 0.00** | **0.04 ± 0.00** | **0.04 ± 0.00** |
| MC Dropout | **0.02 ± 0.00** | 0.06 ± 0.00 | 0.10 ± 0.00 | 0.13 ± 0.00 | 0.17 ± 0.00 | 0.24 ± 0.00 |
| MC Dropout fVI | 0.08 ± 0.00 | **0.06 ± 0.00** | **0.02 ± 0.00** | **0.01 ± 0.00** | 0.05 ± 0.00 | 0.10 ± 0.00 |
| Ensemble | **0.05 ± 0.00** | **0.04 ± 0.00** | **0.02 ± 0.00** | **0.02 ± 0.00** | **0.04 ± 0.00** | 0.10 ± 0.00 |
| Ensemble fVI | 0.18 ± 0.00 | 0.18 ± 0.00 | 0.16 ± 0.00 | 0.14 ± 0.00 | 0.10 ± 0.00 | **0.04 ± 0.00** |
| Radial | **0.09 ± 0.00** | 0.14 ± 0.00 | 0.19 ± 0.00 | 0.23 ± 0.00 | 0.27 ± 0.00 | 0.34 ± 0.00 |
| Radial fVI | 0.13 ± 0.00 | **0.12 ± 0.00** | **0.10 ± 0.00** | **0.08 ± 0.00** | **0.05 ± 0.00** | **0.04 ± 0.00** |
| Rank1 | 0.16 ± 0.00 | 0.22 ± 0.00 | 0.28 ± 0.00 | 0.32 ± 0.00 | 0.37 ± 0.00 | 0.44 ± 0.00 |
| Rank1 fVI | **0.13 ± 0.00** | **0.13 ± 0.00** | **0.11 ± 0.00** | **0.09 ± 0.00** | **0.06 ± 0.00** | **0.03 ± 0.00** |

Table 20: Accuracies for the CIFAR100 adversarial attack experiment. Means and standard errors over ten seeds. Best results within archetype in boldface, best results overall in blue.

| CIFAR100 | Adversarial Attack Epsilon | | | | | | |
|---|---|---|---|---|---|---|---|
| Accuracy ↑ | 0.00 | 0.05 | 0.10 | 0.15 | 0.20 | 0.25 | 0.30 |
| MAP | $\mathbf{75.68 \pm 0.07}$ | $11.84 \pm 0.13$ | $6.70 \pm 0.08$ | $4.98 \pm 0.08$ | $4.18 \pm 0.07$ | $3.62 \pm 0.09$ | $3.18 \pm 0.09$ |
| MAP fVI | $74.77 \pm 0.09$ | $\mathbf{15.83 \pm 0.11}$ | $\mathbf{9.79 \pm 0.12}$ | $\mathbf{7.08 \pm 0.09}$ | $\mathbf{5.38 \pm 0.05}$ | $\mathbf{4.30 \pm 0.08}$ | $\mathbf{3.46 \pm 0.07}$ |
| MC Dropout | $\mathbf{74.05 \pm 0.08}$ | $19.55 \pm 0.10$ | $10.56 \pm 0.08$ | $7.45 \pm 0.09$ | $5.90 \pm 0.09$ | $4.91 \pm 0.06$ | $4.01 \pm 0.07$ |
| MC Dropout fVI | $71.61 \pm 0.10$ | $\mathbf{21.60 \pm 0.07}$ | $\mathbf{13.22 \pm 0.09}$ | $\mathbf{9.52 \pm 0.09}$ | $\textcolor{blue}{\mathbf{7.19 \pm 0.12}}$ | $\textcolor{blue}{\mathbf{5.62 \pm 0.11}}$ | $\textcolor{blue}{\mathbf{4.49 \pm 0.11}}$ |
| Ensemble | $\textcolor{blue}{\mathbf{79.13 \pm 0.05}}$ | $26.04 \pm 0.05$ | $13.51 \pm 0.09$ | $8.64 \pm 0.07$ | $6.32 \pm 0.06$ | $\mathbf{4.92 \pm 0.07}$ | $\mathbf{3.97 \pm 0.07}$ |
| Ensemble fVI | $75.89 \pm 0.06$ | $\textcolor{blue}{\mathbf{29.32 \pm 0.06}}$ | $\textcolor{blue}{\mathbf{16.00 \pm 0.08}}$ | $\textcolor{blue}{\mathbf{10.09 \pm 0.07}}$ | $\mathbf{6.77 \pm 0.05}$ | $4.82 \pm 0.05$ | $3.66 \pm 0.06$ |
| Radial | $\mathbf{76.42 \pm 0.08}$ | $12.05 \pm 0.08$ | $7.36 \pm 0.06$ | $5.57 \pm 0.05$ | $4.52 \pm 0.07$ | $3.77 \pm 0.07$ | $3.19 \pm 0.08$ |
| Radial fVI | $75.29 \pm 0.10$ | $\mathbf{16.85 \pm 0.12}$ | $\mathbf{10.97 \pm 0.09}$ | $\mathbf{7.91 \pm 0.09}$ | $\mathbf{6.06 \pm 0.10}$ | $\mathbf{4.83 \pm 0.12}$ | $\mathbf{3.99 \pm 0.10}$ |
| Rank1 | $73.76 \pm 0.07$ | $14.01 \pm 0.09$ | $9.54 \pm 0.08$ | $7.63 \pm 0.12$ | $6.18 \pm 0.12$ | $5.09 \pm 0.14$ | $4.20 \pm 0.13$ |
| Rank1 fVI | $\mathbf{75.58 \pm 0.09}$ | $\mathbf{18.34 \pm 0.08}$ | $\mathbf{11.68 \pm 0.07}$ | $\mathbf{8.65 \pm 0.10}$ | $\mathbf{6.67 \pm 0.09}$ | $\mathbf{5.29 \pm 0.08}$ | $\mathbf{4.38 \pm 0.08}$ |

Table 21: Log-likelihoods for the CIFAR100 adversarial attack experiment. Means and standard errors over seeds. Best results within archetype in boldface, best results overall in blue.

| CIFAR100 | Adversarial Attack Epsilon | | | | | | |
|---|---|---|---|---|---|---|---|
| Log-Likelihood ↑ | 0.00 | 0.05 | 0.10 | 0.15 | 0.20 | 0.25 | 0.30 |
| MAP | $\mathbf{-1.00 \pm 0.00}$ | $-6.64 \pm 0.03$ | $-6.72 \pm 0.03$ | $-6.53 \pm 0.03$ | $-6.52 \pm 0.05$ | $-6.64 \pm 0.06$ | $-6.76 \pm 0.08$ |
| MAP fVI | $-1.20 \pm 0.00$ | $\mathbf{-5.36 \pm 0.01}$ | $\mathbf{-5.62 \pm 0.02}$ | $\mathbf{-5.53 \pm 0.02}$ | $\mathbf{-5.45 \pm 0.02}$ | $\mathbf{-5.40 \pm 0.02}$ | $\mathbf{-5.36 \pm 0.02}$ |
| MC Dropout | $\mathbf{-0.97 \pm 0.00}$ | $-5.02 \pm 0.01$ | $-5.93 \pm 0.02$ | $-6.08 \pm 0.02$ | $-6.23 \pm 0.02$ | $-6.46 \pm 0.03$ | $-6.71 \pm 0.05$ |
| MC Dropout fVI | $-1.17 \pm 0.00$ | $\mathbf{-4.36 \pm 0.01}$ | $\mathbf{-5.17 \pm 0.02}$ | $\mathbf{-5.35 \pm 0.01}$ | $\mathbf{-5.45 \pm 0.01}$ | $\mathbf{-5.55 \pm 0.01}$ | $\mathbf{-5.66 \pm 0.02}$ |
| Ensemble | $\textcolor{blue}{\mathbf{-0.81 \pm 0.00}}$ | $\textcolor{blue}{\mathbf{-3.68 \pm 0.00}}$ | $\textcolor{blue}{\mathbf{-4.41 \pm 0.00}}$ | $\textcolor{blue}{\mathbf{-4.65 \pm 0.01}}$ | $\textcolor{blue}{\mathbf{-4.81 \pm 0.01}}$ | $\textcolor{blue}{\mathbf{-4.93 \pm 0.02}}$ | $\textcolor{blue}{\mathbf{-5.01 \pm 0.02}}$ |
| Ensemble fVI | $-1.18 \pm 0.00$ | $-3.97 \pm 0.00$ | $-4.86 \pm 0.01$ | $-5.10 \pm 0.01$ | $-5.15 \pm 0.01$ | $-5.17 \pm 0.01$ | $-5.18 \pm 0.01$ |
| Radial | $\mathbf{-0.98 \pm 0.00}$ | $-6.73 \pm 0.02$ | $-6.80 \pm 0.03$ | $-6.69 \pm 0.03$ | $-6.79 \pm 0.05$ | $-7.04 \pm 0.07$ | $-7.28 \pm 0.10$ |
| Radial fVI | $-1.21 \pm 0.00$ | $\mathbf{-5.14 \pm 0.01}$ | $\mathbf{-5.36 \pm 0.02}$ | $\mathbf{-5.29 \pm 0.02}$ | $\mathbf{-5.22 \pm 0.02}$ | $\mathbf{-5.18 \pm 0.02}$ | $\mathbf{-5.16 \pm 0.02}$ |
| Rank1 | $-1.48 \pm 0.00$ | $-9.99 \pm 0.02$ | $-10.69 \pm 0.02$ | $-10.76 \pm 0.02$ | $-10.81 \pm 0.04$ | $-10.93 \pm 0.05$ | $-11.12 \pm 0.07$ |
| Rank1 fVI | $\mathbf{-1.17 \pm 0.00}$ | $\mathbf{-4.95 \pm 0.01}$ | $\mathbf{-5.26 \pm 0.01}$ | $\mathbf{-5.22 \pm 0.01}$ | $\mathbf{-5.17 \pm 0.01}$ | $\mathbf{-5.16 \pm 0.01}$ | $\mathbf{-5.16 \pm 0.02}$ |

Table 22: Expected calibration errors for the CIFAR100 adversarial attack experiment. Means and standard errors over ten seeds. Best results within archetype in boldface, best results overall in blue.

| CIFAR100 | Adversarial Attack Epsilon | | | | | | |
|---|---|---|---|---|---|---|---|
| ECE ↓ | 0.00 | 0.05 | 0.10 | 0.15 | 0.20 | 0.25 | 0.30 |
| MAP | $\mathbf{0.08 \pm 0.00}$ | $0.55 \pm 0.00$ | $0.52 \pm 0.00$ | $0.49 \pm 0.00$ | $0.49 \pm 0.01$ | $0.49 \pm 0.01$ | $0.51 \pm 0.01$ |
| MAP fVI | $0.12 \pm 0.00$ | $\mathbf{0.23 \pm 0.00}$ | $\mathbf{0.23 \pm 0.00}$ | $\mathbf{0.22 \pm 0.00}$ | $\mathbf{0.23 \pm 0.00}$ | $\mathbf{0.23 \pm 0.01}$ | $\mathbf{0.23 \pm 0.01}$ |
| MC Dropout | $\textcolor{blue}{\mathbf{0.03 \pm 0.00}}$ | $0.45 \pm 0.00$ | $0.47 \pm 0.00$ | $0.44 \pm 0.00$ | $0.43 \pm 0.00$ | $0.43 \pm 0.00$ | $0.44 \pm 0.01$ |
| MC Dropout fVI | $0.09 \pm 0.00$ | $0.26 \pm 0.00$ | $0.29 \pm 0.00$ | $0.28 \pm 0.00$ | $0.28 \pm 0.00$ | $0.29 \pm 0.00$ | $0.30 \pm 0.00$ |
| Ensemble | $\mathbf{0.05 \pm 0.00}$ | $0.26 \pm 0.00$ | $0.31 \pm 0.00$ | $0.30 \pm 0.00$ | $0.30 \pm 0.00$ | $0.30 \pm 0.01$ | $0.29 \pm 0.01$ |
| Ensemble fVI | $0.18 \pm 0.00$ | $\textcolor{blue}{\mathbf{0.08 \pm 0.00}}$ | $\textcolor{blue}{\mathbf{0.14 \pm 0.00}}$ | $\textcolor{blue}{\mathbf{0.15 \pm 0.00}}$ | $\textcolor{blue}{\mathbf{0.16 \pm 0.00}}$ | $\textcolor{blue}{\mathbf{0.17 \pm 0.00}}$ | $\textcolor{blue}{\mathbf{0.18 \pm 0.00}}$ |
| Radial | $\mathbf{0.09 \pm 0.00}$ | $0.57 \pm 0.00$ | $0.54 \pm 0.00$ | $0.51 \pm 0.00$ | $0.51 \pm 0.01$ | $0.53 \pm 0.01$ | $0.55 \pm 0.01$ |
| Radial fVI | $0.13 \pm 0.00$ | $\mathbf{0.21 \pm 0.00}$ | $\mathbf{0.21 \pm 0.00}$ | $\mathbf{0.20 \pm 0.00}$ | $\mathbf{0.20 \pm 0.00}$ | $\mathbf{0.20 \pm 0.01}$ | $\mathbf{0.19 \pm 0.01}$ |
| Rank1 | $0.16 \pm 0.00$ | $0.76 \pm 0.00$ | $0.76 \pm 0.00$ | $0.73 \pm 0.00$ | $0.71 \pm 0.00$ | $0.71 \pm 0.00$ | $0.70 \pm 0.01$ |
| Rank1 fVI | $\mathbf{0.13 \pm 0.00}$ | $\mathbf{0.20 \pm 0.00}$ | $\mathbf{0.21 \pm 0.00}$ | $\mathbf{0.20 \pm 0.00}$ | $\mathbf{0.20 \pm 0.00}$ | $\mathbf{0.20 \pm 0.00}$ | $\mathbf{0.20 \pm 0.01}$ |

# H ABLATION STUDIES

## H.1 SAMPLES DURING TRAINING

In Section 3.3, a Dirichlet estimation procedure was proposed using $M$ samples. In the single sample case $M = 1$, motivated by MAP models, a crude approximation was proposed to approximate the precision with the number of training data samples.

For many VIPs used in BNNs, the computational cost of sampling during training means that only 1 sample is used in practice. This means that the 1 sample approximation is used for more than just MAP, but also for MC Dropout, Radial and Rank1 BNNs as well in practice.

To assess the consequence of this approximation, we repeated the CIFAR10 corruption experiment for MC dropout, but with 5 samples to reflect that of ensembles. Figure 10 shows the result. It can be seen that the 5 sample MC Dropout performance is closer to the 1 sample MC Dropout performance than the Ensemble. This result indicates that the VIP model, rather than samples-during-training, has greater impact. The similarity in performance between 1 and 5 sample MC Dropout suggests the 1 sample approximation is reasonable for the training time savings.

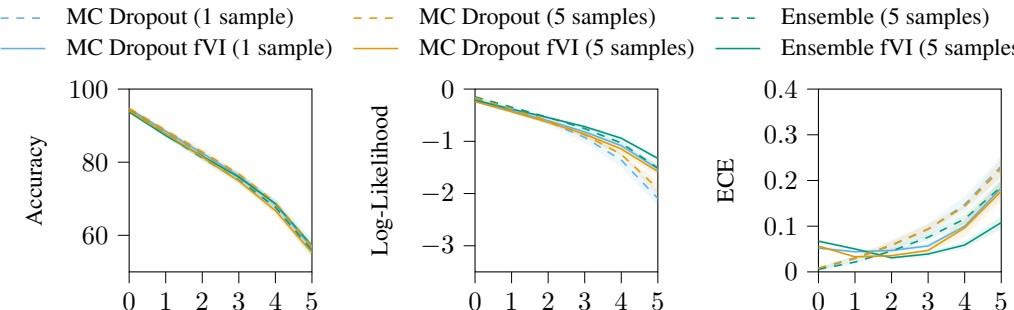

Figure 10: Reproduction of CIFAR10 corruption results in Figure 5, including MC Dropout results with 5 predictive samples during training. For 5 sample MC Dropout, 3 random seeds were used rather than 10 due to the additional training time.

## H.2 SCALING ISSUES WITH HIGH LABEL DIMENSIONALITY

The CIFAR100 experiments revealed an issue with the fELBO objective that caused underfitting for the larger label dimension. To examine why, recall that the categorical likelihood is $\log f_{\boldsymbol{x}\,k}$ when $y_k = 1$. Therefore, the dimensionality of $\boldsymbol{y}$ does not directly influence the value. Conversely, the KL divergence between two Dirichlet densities $D_{\mathrm{KL}}(p_1||p_2)$ does incorporate the label dimensionality $K$ (Rauber et al., 2008),

$$D_{\mathrm{KL}}(p_1||p_2) = \log\Gamma(z^{(1)}) - \sum_{k=1}^{K}\log\Gamma(\alpha_k^{(1)}) - \log\Gamma(z^{(2)}) + \sum_{k=1}^{K}\log\Gamma(\alpha_k^{(2)})$$

$$+ \sum_{k=1}^{K}(\alpha_k^{(1)} - \alpha_k^{(2)})(\Psi(\alpha_k^{(1)}) - \Psi(\alpha_k^{(2)})).$$

To counteract this linear increase due to the summation terms, we can assess a heuristic annealing scale factor on the fKL during training of $1/K = \lambda$,

$$\mathcal{L}(\boldsymbol{\theta}) = \mathbb{E}_{\boldsymbol{f}\sim q(\cdot|\boldsymbol{\theta})}\left[\log p(\mathcal{D}|\boldsymbol{f})\right] - \lambda D_{\mathrm{KL}}[q(\boldsymbol{f}|\boldsymbol{\theta})\,||\,p(\boldsymbol{f})].$$

To investigate this relationship between the Dirichlet KL divergence and the number of classes of the classification, we conducted a toy experiment in a hypercube $[-1,1]^D$ with fixed number of input dimensions $D$ and increasing number of classes $K$. The classes were created by using each dimension as the decision boundary, i.e. $\boldsymbol{x}_d = 0$, and leveraging all permutations to create up to $K = 2^D$ classes, where $D$ was set to 8. The training data, index set for fKL, and test data, consisting of 1000 data points each, were all sampled uniformly at random from the hypercube. We used a MAP model with 2-layer MLP architecture with 25 hidden units each, bias terms enabled, and ReLU activation functions. In terms of optimization, we trained for 3000 epochs using Adam optimizer with a learning rate of 0.005 and default parameters otherwise.

Figure 11 illustrates the model's test log-likelihood and fKL after training. The regular MAP model represents a decent baseline with linear decrease in performance as the label dimensionality increases. In contrast, the test log-likelihood of the MAP fVI model without KL scaling decreases exponentially while the fKL increases approximately linearly. Applying the above proposed scaling to the fKL during training aids the optimization, keeps the final fKL at convergence consistently low and significantly improves the model's test log-likelihood.

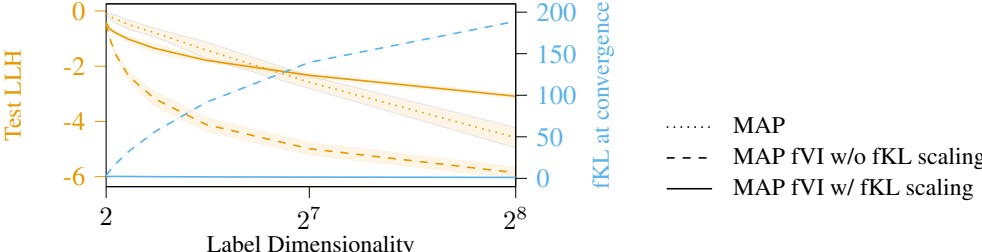

Figure 11: A toy classification example on a hypercube to illustrate the scaling issues associated with the Dirichlet density. The vanilla MAP performance acts as a baseline, and demonstrates the typical range of log likelihood values for this task across increasing label dimensionality. For fVI, the Dirichlet fKL reports a significantly larger range that is x100 the log likelihood range. This value imbalance affects the fELBO objective, resulting in significant underfitting. Applying a heuristic to scale the fKL term, keeping the fKL invariant across label dimensionality avoids the underfitting phemonema. Plot reports mean and 2 standard deviations over 10 seeds.

### H.3 Changing prior Dirichlet parameters

The uniform Dirichlet distribution with concentration parameters $\beta_k = 1$ is a natural choice for an uninformed prior over the simplex. However, potentially interesting cases to consider are priors where all $\beta_k$ are set to another value which is greater or smaller than 1. While the Dirichlet mean remains the same, $\beta_k > 1$ corresponds to greater confidence that the class probabilities are uniformly distributed and $\beta_k < 1$ prefers dominance of any particular class. It was also hypothesized that scaling $\beta_k$ could yield results comparable to scaling the fKL as discussed in the previous subsection. To test this hypothesis, we repeated the hypercube experiment from the previous subsection with the MAP fVI model without fKL scaling while using different $\beta_k$ as prior parameters.

Figure 12 show the test log-likelihood of the MAP fVI model without fKL scaling after training with Dirichlet priors using varying prior concentration parameters $\beta_k$. However, there are no significant differences when using different $\beta_k$ and no particular $\beta_k$ achieves test log-likelihoods which would be comparable to the improvements due to fKL scaling discussed in the previous subsection.

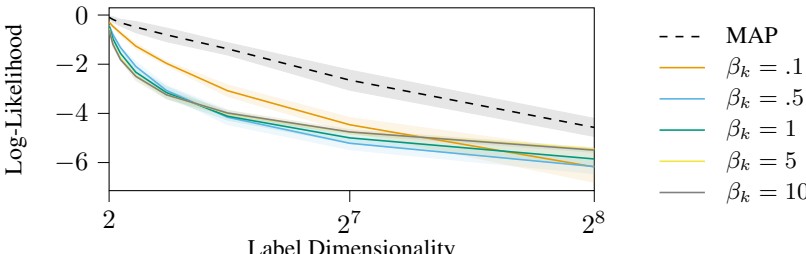

Figure 12: Insert caption.

To further investigate the Dirichlet prior with different concentration parameters, we repeated the visualizable Two Moons toy problem using the MC Dropout fVI model with different $\beta_k$. Figure 13 depicts the predicted class probabilities of the toy problem with $K = 2$ classes. For $\beta_k < 1$, the areas of confident prediction enlarge but quickly fall back to uniformity, whereas for $\beta_k > 1$, the confident predictions or more locally concentrated, tapering slowly towards uniformity and 'leaking' into the unobserved area.

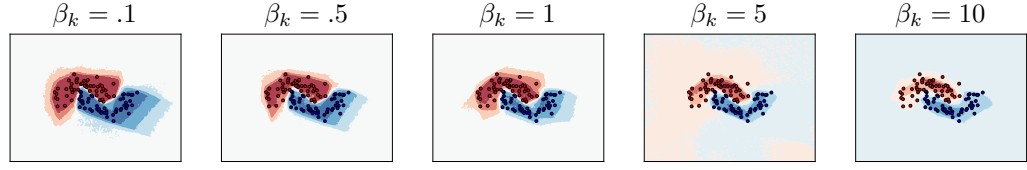

Figure 13: Reproduction of the Two Moons toy problem (Figure 3) with varying uniform prior precision.

