# OpenReview forum: "Function-Space Variational Inference for Deep Bayesian Classification"
_ICLR.cc/2022/Conference — ICLR 2022 Submitted_

### Official Review · Reviewer_rF5E · 2021-11-01

**Correctness:** 2
**Technical Novelty And Significance:** 2
**Empirical Novelty And Significance:** 2
**Recommendation:** 3
**Confidence:** 4

**Main Review:**

Summary:

See "Summary Of The Paper".

---

Reasons for score:

The idea is interesting and the paper writing is clear. However, my major concern is about the theoretical soundness of the proposed method. See my detailed comments below.

---

Pros:

- The paper is well-organized and easy to follow.
- The discussion on related work is comprehensive and systematic.
- Substantial experimental results are provided on several different important tasks (especially in the Appendix).

---

Concerns:

My main concern is about the theoretical soundness of the proposed method. In Sec. 3.2 Eq. (4), the author mentioned that _a more tractable approximation replace the supermum with an expectation_, $D_{KL}[q(f|\theta)\|p(f)]\approx E_{s\sim S} D_{KL}[q(f_s|\theta)\|p(f_s)]$. However, in the original fVI method (Sun et al., 2019), the sampling-based method samples **measurement sets**, i.e., $D_{KL}[q(f|\theta)\|p(f)]\approx E_{S\sim c} D_{KL}[q(f_S|\theta)\|p(f_S)]$, rather than **data points in the measurement set**. Note that there is a crucial difference: we will be estimating the KL divergence between the *joint* distributions induced by the variational and the prior if we sample measurement sets. In Eq. (4), however, the dependence among predictions (in both prior and variational distribution) at different input $x$ is completely ignored. This perhaps leads to meaningless results since dependence is the core of functions. Example:

Suppose that in a 10-class classification problem, we have 1000 data points located nearly at $x_A$, and the class labels are evenly distributed (that is, 100 data with label 1, 100 data with label 2,..., 100 data with label 10). Assume we also have 10 data points located at $x_B$ (1 data with label 1, 1 data with label 2,..., 1 data with label 10). Intuitively the ideal posterior predictive Dirichlet mean at $x_A$ and $x_B$ should be $(1/10, \cdots, 1/10)$, but the predictive Dirichlet precision should be different since we see more data at $x_A$. However, according to Eq. (1) and Eq. (6), the number of likelihood terms and the KL term are always equal at both $x_A$ and $x_B$, leading to equal posterior predictive Dirichlet distributions.


My second concern is about Sec. 3.3. I'm not sure whether it makes sense to estimate the distributions of $f^{(m)}_x$ with Dirichlet distribution. In the last line of Page 3, the author mentioned that *sampling the weights and evaluating $g(x,\cdot)$ is equivalent to sampling the Dirichlet predictive*. Is there any intuition that predictive distribution induced by the variational implicit process is approximately Dirichlet?





**Summary Of The Paper:**

This paper proposes a function-space variational inference method for classification tasks. The method is based on the recently proposed functional variational inference (fVI), which directly approximates the posterior over functions instead of the model parameters. The authors use a variational implicit process as a parameterized posterior and adopt Dirichlet predictive priors for the categorical predictive distributions. Since the predictive posterior is defined implicitly, the authors further propose an iterative approach to estimate the Dirichlet predictive posterior, and end up with a wake-sleep style inference procedure. Finally, the proposed method is verified on several standard classification tasks.

**Summary Of The Review:**

In summary, I raise several concerns about the theoretical soundness of the proposed method, which currently prevents me from rating this paper higher. I will consider raising my evaluation if the authors address my concerns or point out my misunderstandings.

---

> ### Author Response · Authors · 2021-11-19
> **Response to Reviewer rF5E**
>
> This review was very helpful, thank you.
>
> **fKL estimation.** You raised an important detail that we did not clarify in the initial submission. Prior work fBNNs had rich GP priors and so evaluating this joint is both possible and meaningful. However, by the very design of the BNNs, the predictive Dirichlet joint over different inputs $x$ factorizes into independent predictions whose dependence is not explicitly modeled using e.g. a GP kernel, but instead implicitly learned by the neural network. While we admit this factorization, similar to mean-field VI in BNN weight space inference, is not a rich prior or posterior, it is theoretically sound. We have fixed the notation throughout and discussed this design choice in the new draft.
>
> In terms of the actual training algorithm, our approach reflects Sun et al.'s procedure. Sun et al. sample a measurement set from a fixed distribution $c$ during each batch iteration, whereas we used a mini-batch from a fixed index set during each batch iteration. This mini-batch can be understood as a sample from an empirical distribution, i.e. the index set, because it is difficult to construct a distribution over natural images. We believe the misunderstanding arose due to confusion with our notation and the terms 'measurement set' and 'index set' , and we have thus changed the notation and added sentences to clarify the respective sections.
>
> Regarding your example, this is a question of how our model represents uncertainty. To represent epistemic uncertainty, each individual prediction should favor a different class (e.g. ensemble disagreement). To represent aleatoric uncertainty, each prediction should be a uniform categorical distribution. In your specific example, the Dirichlet posterior mean, i.e. categorical prediction, would be uniform at A and B because it has seen uniform data there, as you have already explained. However, the precision would depend on the (dis)agreement of the individual M predictions. So, we could ask ourselves whether e.g. a MC Dropout model would produce a greater variety of predictions at B than A. If a greater variety of categorical predictions is produced at B then those M disagreeing predictions would result in a low posterior Dirichlet precision via Minka’s MLE algorithm. If the variety of predictions is the same for A and B then the resulting precision would also be the same at A and B. Therefore, we are leveraging but also bottlenecked by the performance of the underlying backbone model. Nonetheless, we argue that our approach is an improvement over the conventional usage of such models, which would only be able to produce the same uniform categorical prediction at A and B even if there were differences in the variety of individual predictions (see Figure 2).
>
> **Dirichlet assumption.** By placing a Dirichlet predictive prior and having a (conjugate) categorical likelihood as an observation model, we obtain a Dirichlet posterior. This is different to whether the BNN output is naturally Dirichlet distributed, which is not guaranteed, but intuitively motivated due to prior conjugacy of categorical and Dirichlet distributions. Hence the Dirichlet distribution is a design choice suited for classification tasks. This choice allows us to influence the implicit predictive distribution via the Dirichlet prior in the fVI objective to improve uncertainty quantification.
>
> We hope this addresses your concerns. Let us know if you have further questions.

---

### Official Review · Reviewer_why2 · 2021-11-02

**Correctness:** 3
**Technical Novelty And Significance:** 2
**Empirical Novelty And Significance:** 2
**Recommendation:** 6
**Confidence:** 3

**Main Review:**

## Strengths

- **Significance.** This is a timely contribution that seeks to address significant limitations of Bayesian deep learning, namely those concerning the calibration of uncertainty and robustness to adversarial attacks.
- **Clarity.** The paper is generally well-written and mostly self-contained. I found it clear and easy to follow. I particularly appreciated the treatment of the prior works related to this paper, which I found to be extensive while also being succinct and clear.
- **Reproducibility.** It's excellent to see the depth of low-level implementation details that are provided in the appendix. This should greatly aid subsequent works in reproducing the results reported in this paper. It was also encouraging to see that the code is included in the supplementary material and that it seems to be clean and well-structured.

## Weaknesses

- **Novelty.** Being a combination of implicit variational processes, functional variational inference, and basic concepts from probabilistic multi-class classification,  the novelty of the contribution is somewhat limited. I find the claim that this work provides "a unifying view of prior work which use the Dirichlet distribution and function-space regularization" to be a slight over-statement. A method that combines concepts from two areas is not necessary a unifying theory thereof. One significant technical challenge that is tackled is the MC estimation and minimization of the KL divergence, which is addressed using a quasi-Newton method.

## Questions

- Section 4.4 considers image classification at five increasing levels of corruption. It is understood that each of these numbers corresponds to a particular setting of brightness, contrast, saturation, etc. However, what these settings are remains unclear. Is this detailed anywhere?
- Do you actually make use of the ability to incorporate class-biased predictive prior that the proposed approach unlocks in any experiments other than the qualitative results shown in Figure 3? More generally, are there concrete problems in which having strong informative predictive prior for multiclass classification is actually helpful, and where conventional Bayesian approaches, through posterior inference with its data-fit component are unable to adapt to the class-biases. It would be compelling to see quantitative results that support the claims concerning class-biased predictive priors.
- I am not fully convinced about the results shown in Section 4.4, in particular in Figure 5 for CIFAR100 where the accuracy and log-likelihoods are not distinctly better across corruption levels, particularly with Ensembles. Furthermore, the ECE is only lower for the highest levels of corruption. The explanation offered is that the method underfits, which is supported by the relatively low log-likelihood and accuracy. Obviously one can easily attain good ECE with OOD data by deliberately underfitting, which is exactly why the log-likelihood and accuracy are important to consider as well. Therefore, I don't understand why the following comment from the authors isn't actually done? "This suggests the function-space prior should be fine-tuned (i.e. empirical Bayes) if superior ECE is desired in-distribution." When would this not be desired?

## Miscellaneous Issues

- Pg. 2: "which use Gaussian weight priors and posteriors" → "that uses [...]"
- Pg. 4: "evidence lower bound (ELBO)" - the ELBO and, more generally, variational inference, long predates Hoffman et al. 2013.
- Pg. 4: "Further details [...], which we consider, [...]" - which → that; extraneous commas
- Pg. 8: "LLH" this abbreviation was never defined.
- The abbreviation "OOD" has even been explicitly defined. Likewise, the abbreviation "ECE" has never been defined, nor is any explanation given as to what it is. Yet it is one of the pivotal metrics considered in this line of work.

**Summary Of The Paper:**

This paper considers a functional variational inference approach to Bayesian deep learning for multi-class classification.  This paradigm enables the use of a Dirichlet predictive prior in function-space, which would otherwise be cumbersome under the classical weight-space treatment. The authors approximate the posterior in function space through the broad family of variational implicit processes, which are able to subsume popular BNN paradigms such as deep ensembles and Monte Carlo dropout. Experiments are provided to support the claim that this combination of function-space regularization with a Dirichlet predictive prior improves adversarial robustness and uncertainty quantification over its weight-space counterparts.

**Summary Of The Review:**

Given the technical quality of the paper, I am leaning toward recommending acceptance of the paper. However, given some concerns around novelty and empirical evaluations detailed in my review, I am disinclined to recommend a clear acceptance.

---

> ### Author Response · Authors · 2021-11-19
> **Response to Reviewer why2**
>
> We thank the reviewer for their kind words about the submission. We hope to answer their concerns and queries.
>
> **Novelty.** We agree that a ‘unifying view’ is indeed quite a strong term. We have changed it to ‘perspective’, as we were motivated by function-space VI but soon realised these works have similar objectives. Our motivation for claiming unification is because each prior work misses certain aspects. Belief matching introduces the ELBO objective but does not design an index set or function-space view. Prior networks use an index set as ‘OOD regularization’ but do not have an ELBO objective and therefore rely on manually designing the weighting and target distributions. Both methods are not fully Bayesian, as they directly predict Dirichlet parameters rather than having a ‘natural’ Dirichlet predictive that you get from VIPs. Therefore our ‘unified’ view shows how this objective can be designed and optimized effectively. However, while the ELBO comes with a natural weighting parameter, in practice we need to anneal it to improve performance so we fall short of fully removing the hyperparameters.
> Please note that our novelty also arises from framing fVI for classification via the Dirichlet and scaling inference to large-scale image classification tasks. This is an area of active research from multiple groups, as previous fVI work has focused on regression tasks and using GP priors.
>
> **Corrupted CIFAR10.** This task is detailed in: Dan Hendrycks and Thomas Dietterich.   Benchmarking neural network robustness to commoncorruptions and perturbations. In International Conference on Learning Representations, 2019.
>
> **Class-biased priors.** As with designing any prior, it ultimately depends on the task and the domain knowledge. Our hope is that there is a community who would find this capability useful. One general application is an empirical Bayes procedure, where you would refine the model by using the old model as the prior for the next model. This comes with training expense, but allows you to overfit more to the training data if your initial prior is too restrictive. We did not implement this scheme for this paper as we were focused on analysing the uniform prior and assessing OOD performance rather than in-distribution performance, and implementing it for the existing codebase would more code changes than the rebuttal time allows, therefore we did not have time to try it out for this rebuttal.
>
> **CIFAR100 underfitting.** We concede that results are less impressive for CIFAR100. However, Figure 7 shows there is still a benefit for most models, although delayed compared to CIFAR10. We were also humbled by the effectiveness of ensembles at this task. Regarding our statement that "this suggests the function-space prior should be fine-tuned (i.e. empirical Bayes) if superior ECE is desired in-distribution": Ultimately there are no free-lunches, and we believe our model's performance is similar to the accuracy-vs-robustness trade-off seen in adversarial robustness. Therefore, there is the risk that improving in-distribution performance sacrifices out-of-distribution performance, which is why improving in-distribution performance may not be desired. We did not pursue empirical Bayes in this submission as we wanted to focus on the capability of the uniform prior, and we see empirical Bayes as future work to further improve the performance of fVI.
>
> We hope this response is to your satisfaction. Let us know if you have further questions.

---

> > ### Comment · Reviewer_why2 · 2021-11-29
> > **Acknowledgement**
> >
> > Many thanks to the authors for providing a response to my concerns. After having followed the discussions on the points raised by the other reviewers and taking these into consideration, I have decided to stick to my original overall score.

---

### Official Review · Reviewer_FSJj · 2021-11-02

**Correctness:** 4
**Technical Novelty And Significance:** 3
**Empirical Novelty And Significance:** 3
**Recommendation:** 6
**Confidence:** 3

**Main Review:**

Firstly, this work is relevant for ICLR: designing function-space priors and inference for Bayesian neural networks is a hot topic right now. I also thought that this paper was written well, with good motivation and explanations throughout.

I like the overall idea of taking multiple outputs from a Bayesian neural network, and instead of integrating them out to get a categorical distribution, using the samples to estimate a Dirichlet distribution (and using Minka's algorithm to estimate the parameters of the Dirichlet).

This work is closely related to prior works such as Joo et al., 2020 and others. Appendix A summarises these works and I thought nicely summarises the differences between this method and these closest prior works. However, although theoretically there are differences between these past works and the overall framework of fVI that this paper introduces, it does not seem to me like there are significant differences in practice on large datasets. More concretely:

1. The authors say that their framework allows for index sets that are not just the training data. However, in all but one of their experiments, only the training data is used. It is nice to see the rotated MNIST example with a larger index set, however, this is quite a limited experiment (as it is only MNIST, and only tested in a very specific OOD regime). I would have liked to see more experiments where the authors used data augmentation or unlabelled data as part of the index set. Or, used some sampling similar to Sun et al., 2019, where they sample from some interval to get data not in the training set. By not having such experiments, the paper becomes similar to past work.

2. The other difference is that fVI allows for Bayesian neural networks with M>1 samples in order to estimate the Dirichlet distribution. I like this idea. However, in practice, I was disappointed that M>1 only for ensembles and Rank1 models, and not also for MC dropout and Radial BNNs.

Other comments / concerns:

3. Why were z_min and z_max chosen as K and N respectively (Appendix F)? And why was z estimated to be N when M=1 (why not estimate z using Minka's algorithm)? From what I understand, as N becomes very large, the distribution would approach the same categorical that people use. N is quite large for many datasets already, so effectively this procedure is adding a very small amount of noise around what would otherwise be the categorical. I understand this reduces training cost. But it would be nice to see results for eg Radial BNNs with larger M, and also see how the estimated z compares with z=N (or z=K). (Appendix H is very short and I would like to see this discussed in more detail, including more in the main text.)

4. fVI versions of algorithms are almost always doing worse in terms of loglik and ECE on the training data. I had hoped for better 'uncertainty' performance on the in-distribution set as well. This indicates to me that the fVI version still over-estimates in-distribution data (eg MAP vs MAP fVI). The real benefit is only coming when the test distribution is not the same as the training distribution. Do the authors have some insight as to why it is not helping in-distribution? In Section 4.4 the authors say, "due to the predictive prior, the ECE was significantly higher for corruption 0". I did not understand this argument, can the authors please elaborate?

5. Did the authors try different prior parameters (rather than beta=1)? It seems like this might affect results considerably, and I would be interested to see this more (I liked the third and fourth columns of Figure 3). For example, on the toy dataset (Figure 3), what happens when using beta = [5,5] or even something like beta = [0.8,0.8]. In fact, in general, I think the ideal solution for Figure 3 for regions outside the input-data-space, would be to *slightly* predict the closest class (eg the top left corner should unconfidently predict red, and the bottom right should unconfidently predict blue). Perhaps different beta would allow for this.

6. At the end of Section 3, it would be good to add a sentence moving some of Appendix B into the main text, by mentioning the authors down-weight the KL term (and that this might be justified by referencing Wenzel et al., 2020 and Aitchison, 2021).

Very minor points:

7. Why is label smoothing applied (Appendix F)? Is this necessary for some reason, eg for numerical stability? Why?

8. Some more work the authors could refer to: (i) Burt et al., 2021, "Understanding VI in function-space", (ii) recent SG-MCMC methods for (weight-space) Bayesian neural networks, (iii) recent natural-gradient variational inference methods for (weight-space) Bayesian neural networks.

**Summary Of The Paper:**

The paper introduces a function-space variational inference algorithm ("fVI") that uses a Dirichlet predictive prior, and approximates the output of a (weight-space-Bayesian) neural network as a Dirichlet distribution. Inference is achieved by viewing the network as a variational impolicit process, and optimising the functional ELBO from Sun et al., 2019. This view allows them to impose priors in function-space to many existing algorithms, such as MAP, MC Dropout, ensembles, Radial BNNs, and so on. The authors show results on a toy problem, and focus on out-of-distribution performance on larger problems such as MNIST, CIFAR10 and CIFAR100, as well as performance under a fast gradient sign method adversarial attack. The fVI variants almost always perform better when the test distribution is not the same as the training distribution.

**Summary Of The Review:**

Overall, I like the approach of the paper, and think there is potential. Therefore, although I am borderline, I am leaning towards accept. My main review outlines my biggest issues (especially points 1-4), these tend to be about the practical implementation not aligning with what I view as the biggest benefits of the theory (and the biggest differences to past works), as well as questions about experimental results.

------------
Post-rebuttal: Please see discussion. I am sticking with my score for now, despite some of my initial concerns remaining (and additional valide points raised by Reviewer rF5E).

---

> ### Author Response · Authors · 2021-11-19
> **Response to Reviewer FSJj (2/2)**
>
> **Z approximation.** For a single sample we cannot estimate the dirichlet precision, therefore we opt for the greatest possible value as an approximation (N) using the intuition that $\alpha$ relates to class pseudocounts (i.e. what happens when you compute the posterior update for a Dirichlet prior). This is also why z_max is set to N, and z_min is set to K because this corresponds to our prior where each alpha is 1. We did not have time this week to expand on Section H.1 on top of the other reviewer requests as we had limited computing resources but we could include results for Radial BNN for a camera-ready version. Given that H.1 already shows little difference for MC Dropout with M=5, we did not see a Radial BNN experiment as a priority. We hope the reviewer understands.
>
> **Underfitting.** The reviewer raises a valid criticism that our models underfit slightly in LLH and ECE on the standard test sets. Given that the belief matching model performs similarly to MAP fVI, we see this as a characterization of function-space inference in this setting, as there is stronger regularization on the predictions. This is caused by the predictive prior encouraging non-zero probability mass on all the other classes for the index set (i.e. training data), which naturally extrapolates to the test data. This regularization doesn’t affect accuracy (note fVI accuracy is sometimes better on the test sets) but does affect LLH and ECE. Therefore, we expect to underfit compared to models that can apply 0 probability mass on other classes when predicting. Given that our fVI approach can yield better performance OOD (with one exception), we interpret this as a trade-off between accuracy and robustness. This trade-off is also seen in the adversarial robustness literature. Of course, ideally we would like to satisfy both in practice, but we see this as future work as it is still an open problem (e.g. A closer look at accuracy vs. robustness. Yang et al. (2020)). We have included more of a discussion on this trade-off in the main text. One possible approach to avoid underfitting is to use empirical Bayes, which would use an old model as a predictive prior rather than our uniform prior. This would result in less regularization on the training data. However, the implication of empirical Bayes (EB) is slower training (as now two models are used) and potentially reduced OOD performance due to overfitting, so there is a trade-off again but we believe EB could still be an interesting extension to improve performance.
>
> **Different priors.** We’ve added a comparison of prior values in the appendix (Figure 12). We saw some variation by changing the precision of the prior while keeping the same mean. $\beta_k = 5$ appears to achieve the fit the reviewer describes.
>
> **Scaling KL.** As we are tight for space we had to move this discussion to the Appendix. We hope that the pointers to the appendix and mentioning the KL scaling in the main text is sufficient for readers to be aware of this property of our method.
>
> **Label smoothing.** This was an imprecise statement on our part. We apply smoothing to the model’s predicted class probabilities for numerical stability, because they are used in log, gamma and digamma functions which are sensitive to small values. The dataset labels are not smoothed. We have fixed this sentence.
>
> **References.** We’ve added Burt et al., ‘Cyclical Stochastic Gradient MCMC for Bayesian Deep Learning’ Zhang et al (2019) and ‘Noisy Natural Gradient as Variational Inference’ Zhang et al (2018). If you have specific references in mind please let us know.
>
> We hope this has satisfied your concerns, but please let us know if you need more clarifications or have further questions.

---

> > ### Comment · Reviewer_FSJj · 2021-11-29
> > **Reviewer response**
> >
> > Thanks to the authors for their response.
> >
> > I have been following the discussion with Reviewer rF5E with interest, and I thank both the reviewer and authors for a lively discussion. I agree with the reviewer's concerns regarding factorised priors. That being said, as the reviewer said, the VIP provides correlated predictive distributions at nearby inputs, causing the posterior to not be factorised. This distinction appears to be very important, and should be made crystal clear in the main paper.
> >
> > The authors' rebuttal have alleviated some of my concerns, and some of the extra plots and writing in the revised paper are nice. I understand that the authors chose to priorities other experiments during the rebuttal phase (this is fine, rebuttal periods do not give much time), however, I still do not see the intuition behind approximating z. It appears to me that the value of z is very important for the success of the algorithm, and it is usually chosen in a heuristic way. Moreover, setting Z to N would effectively mean a very tight Dirichlet distribution (as N is usually very large), making me doubt a key benefit of having a Dirichlet distribution as opposed to just the mean of the samples. The fact that results are very similar for MC Dropout when M=1 and M=5 seems concerning to me, indicating that the extra noise added by the Dirichlet is not very important.
> >
> > I am thinking of sticking with my recommendation of accept for now. This is because the topic is highly relevant for ICLR, and I think the community would overall benefit from seeing this paper right now, as opposed to later. But there are clear ways the paper could improve.

---

> > > ### Author Response · Authors · 2021-11-29
> > > **Author Response**
> > >
> > > We thank the reviewer for their response, understanding, and kind words.
> > >
> > > To answer the remaing concerns
> > >
> > > > I still do not see the intuition behind approximating z. It appears to me that the value of z is very important for the success of the algorithm, and it is usually chosen in a heuristic way.
> > >
> > > The underlying issue here is using a single sample to estimate the predictive mean during training (to reduce training time), which is not unique to our approach and is used across Bayesian deep learning. With only a single sample, $z$ must be approximated heuristically. We chose a heuristic approach that worked in practice. We believe it is not unreasonable to assume that the model has confident predictions when the measurement set is the training data, which is the case for our large-scale experiments where $N$ is also large.
> > >
> > > > ... making me doubt a key benefit of having a Dirichlet distribution as opposed to just the mean of the samples.
> > >
> > > As the mean of our Dirichlet is equal to the sample mean, these two approaches are not so different in practice.
> > >
> > > > The fact that results are very similar for MC Dropout when M=1 and M=5 seems concerning to me, indicating that the extra noise added by the Dirichlet is not very important.
> > >
> > > As the Dirichlet acts as a variational approximation of the predictive distribution, it may just be that MC Dropout ResNet models do not have very diverse predictive samples for CIFAR10 + chosen measurement set, which is independent of our motivation for using a Dirichlet predictive for our inference strategy. We can look into predictive sample diversity as a possible diagnostic metric for our models. The predictive diversity of ensembles has already been looked at [1].
> > >
> > > [1] Deep Ensembles: A Loss Landscape Perspective, Fort et al.

---

> ### Author Response · Authors · 2021-11-19
> **Response to Reviewer FSJj (1/2)**
>
> We’re glad the reviewer enjoyed the paper, and they raise some important points about the submission.
>
> **Novelty w.r.t. prior work.** As shown in Figure 12, the ‘belief matching’ (BM) method of Joo et al. is spiritually similar to our MAP fVI model. The contribution of this work is combining Dirichlet-based fVI with stochastic NNs with implicit predictive distributions. Since the BM approach directly predicts the $\alpha$ parameters, it’s not a ‘true’ BNN and is limited to a deterministic network. Our approach lets us harness the performance of a variety of BNNs like ensembles which demonstrate superior performance to the MAP baseline.  Moreover, while Joo et al. coin a new ‘belief matching’ approach, we show how it is in fact a variation of fVI which better motivates applying a KL to the predictive distribution and introduces the design of index sets.
>
> **Index set design.** This is indeed an important and interesting aspect of fVI. Our toy example in Figure 3 uses the 2D Euclidean space as index set, similar to the uniform distributions in Sun et al. However, this work was primarily motivated to scale fVI to larger-scale, image-based classification tasks. In this setting, designing an index set is harder now because the data domain consists of natural images, so uniform or Gaussian noise does not necessarily generate interesting OOD data. Moreover, in scaling to larger models and tasks, we now have the added complexity of the ResNet CNN hypothesis space and training tricks such as batch norm and data augmentation. To see why this is important, refer to the rotated MNIST task in figure 4. For the MLP in this task, the performance drops below the prior at around 60 degrees (left subfigure). Therefore improving the quality of the fVI with the richer index set improves performance by bringing the model closer to the prior (right subfigure). If you compare this to the corrupted CIFAR10 task (Figure 5), because there are 10 classes again, the prior performance is around -2.3 in terms of log-likelihood. However, presumably due to the inductive bias of CNNs and the data augmentation, the model is better than the prior in the OOD region under evaluation. Therefore, there is not the same motivation for incorporating a richer index set. This is why we left index set design for future work, as we require tasks, priors and OOD data where a richer index set is necessary to improve downstream performance. Moreover, there is the added practical complication of working with batch norm, as it is required for training ResNets, but naively adding OOD data will alter the batch norm statistics which may affect performance. Therefore, a second open question is how to work with both a richer index set and batch norm without affecting performance. A third question is designing an index set for the adversarial robustness task, as now the ‘OOD data’ depends on the model and so the optimization problem (i.e. fELBO) becomes dynamic.
> We believe these studies are interesting investigations in their own right, rather than content for an appendix, therefore we see it as future work for improving large scale fVI performance. However, we have added the uniform prior performance to all plots so this performance difference is clearer.
>
> **Predictive samples.** Note that our approach facilitates M>1 for any implicit model, but in practice this slows down training considerably so a single sample is used. Radial BNNs and MC Dropout and Rank1 BNNs also use 1 predictive sample during training in their papers for this reason. In Section H.1 of the Appendix you can find a small ablation with MC Dropout we did to show that M=5 is reasonably close to M=1, so this does not seem to be a serious approximation w.r.t performance.

---

### Official Review · Reviewer_7nPR · 2021-11-03

**Correctness:** 3
**Technical Novelty And Significance:** 3
**Empirical Novelty And Significance:** 2
**Recommendation:** 5
**Confidence:** 4

**Main Review:**

Pros:

1. The paper is rigorously written, scholarship and attribution of related work is generally good, and the mathematical definitions for all used concepts and background material are abundantly available in the main paper and the appendix. This is quality material, I congratulate the authors for doing the work.

2. The key idea ,while very close to the space of papers described in appendix Section A, is a useful and interesting extension to try to utilize Bayesian estimates of the forward passes to obtain the Dirichlet parameter. While this is a narrow extension and does not introduce a better model, it is technically well justified and distinct as an inference approach.

3. In most cases the approach also seems to be empirically beneficial.

4. The authors have an unusual approach to implicitly estimate the quantity of interest using a nested Newton step, I found this quite interesting and appreciate the link to Tom Minka's old note on the Dirichlet.

Cons:
1. Primarily I want to note that the paper did not need and does not really benefit from the implicit processes connection. The key trick might actually be easier to convey and explain without that distraction in order to present the paper better. It would have been sufficient to mention in the related work section that this can be seen as an instance of this framework. Occasionally clarity suffers when trying to blend too many ideas into one paper. This paper here basically needs fVI and the rest can be derived and explained fine, Definition 3.1 does not seem to carry its weight in terms of prominence to help the reader understand the material.

2. In the otherwise well-designed experiments, I would like to see a clear demonstration of the performance of the models described in appendix Section A. It is valuable and interesting that these models have been explained as special cases of this framework here, but I would still like to see which performance can be achieved with them vis-a-vis with the fukll Bayesian estimator propsoed here to get a full idea of rthe progress this framework offers. As the evaluation currently stands, I understand that this works compared to weight space VI, but not how it works compared to the methods in Sec. A with various Bayesian models. As such, it is hard to evaluate how much benefit the novelty of this paper offers compared to the existing evidential learning-style models if the key trick with the inner Newton step to estimate the Dirichlet parameter were not applied, but for instance the models still were used in a Bayesian way.

3.
In the experiments, the authors repeatedly point out that in CIFAR 100 weight based models do better due to the strong regularisation effect of the Dirichlet prior used here when having this many classes. I appreciate that insight and hypothesis, but could we please test it? How would the performance in this case differ if the prior were to be set to be weaker (but still uniform), i.e. 1/lambda in order to be less informative. According to the authors' arguments, this should improve performance. Could we test that?

4. The authors mention that weight priors cannot obtain estimates of OOD such as the ones shown in Figure 3 middle and right. I would argue that in Hierarchical Gaussian Process Priors For Neural Network Weights by Karaletsos & Bui in Neurips 2020 explicit weight priors were used with similar results, with the key insight that these priors were data-dependent and the sampled weights as such local variables to each datapoint. There is an interesting relationship to this property here, as the Dirichlet estimate also is local to each datapoint. So maybe a softer statement could be that global weight priors may struggle with such generalizations as opposed to local priors?

5. The authors have a mathematical hack in their method which they acknowledge: they disconnect the gradient when separately performing their inner optimization for the Dirichlet parameter and as such do not directly optimize the joint objective. I would be curious to see how the method performs if that gradient were to be carried forward using bi-level optimization (here's a review https://arxiv.org/abs/2101.11517#), as in the current inference we probably do not see the method evaluated fully. I appreciate that the authors claim it still works, but it would be instructive to get a better sense of this.



**Summary Of The Paper:**

The authors propose Function space variational inference for deep Bayesian classification, an inference framework which extends ideas around evidential learning for classification by applying function space Variational Inference and marginalizing over latent functions.
Concretely, this leads to a method where the estimated Dirichlet is not directly regressed from a forward pass of a function, but is estimated as a parameter that would match the empirical distribution over sampled logits given multiple samples from the function posterior (using a nested optimization step), and hence can be plugged into the fVI framework.
The authors show this is a flexible way to model uncertainty for classification and can be combined with diverse models (ensembles, MCDropout and various Bayesian models).
The method produces good results.

**Summary Of The Review:**

I like the premise of this paper and it is clear that this is a work of depth and quality. However, I feel there are some details missing that I would hope can be fixed to make the work clearer and to address some of its weaknesses.
If the authors can address the points I made about the paper (probably the first is hard to do now given the narrative of the manuscript, but the rest may be feasible) and the method is clarified to have clearly identifiable empirical benefits over the baselines in Sec. A, I would be happy to revise my opinion positively and strongly support publication.

Update post-review:
I thank the authors for their measured responses. My sense is this work could still benefit from some more iterations, such as the disconnected gradient and the somewhat dissatisfactory empirical comparisons. As such, while I still like the paper, I cannot clearly recommend acceptance because I would have these questions open as it stands. I suggest working on fixing these aspects to have a more complete version of this manuscript and hope to see it again as a reviewer when that is the case so I can strongly support it.

---

> ### Author Response · Authors · 2021-11-19
> **Response to Reviewer 7nPR**
>
> We thank the reviewer for their kind words and feedback for this work.
>
> 1. **On implicit processes.** It is a fair opinion that this VIP is not necessary to describe our approach. Indeed, the first version did not have this narrative. However, we found that readers did not understand a) how we used BNNs as a variational distribution over functions, and b) how we were doing function-space inference on a finite set of parameters. The VIP definition corresponds exactly to how we used our BNNs, so we incorporated Definition 3.1 and Table 1 to explain the different parametric weight distributions combined with an architecture $\mathbf{g}$ to define an implicit distribution over functions. While this may not be necessary, we believe it is very helpful in making our approach clear and concrete, while also connecting it to related prior work. Moreover, as the VIP references classification models as future work, we see this submission as a necessary extension.
> 2. **Section A models comparison.** We have used the open-source released code of belief matching and prior networks to provide this comparison on corrupted cifar10. As you can see, belief matching is equivalent to our MAP fvi model. With the prior network code, we were unable to reproduce their results with their code in our equivalent setting and the authors did not respond to our emails regarding reproduction. Regarding novelty, belief matching and prior networks used deterministic models to directly predict their uncertainty via Dirichlet parameters. In our work, we are using a Dirichlet predictive prior to influence the predictive uncertainty of the stochastic network in a Bayesian fashion (VI). So while the objectives look similar, the models we use are a crucial difference. The inner Newton step facilitates achieving the fELBO objective with an implicit predictive distribution.
> 3. **High dimensional classes.** We discuss this phenomenon in more depth in Section H.2. Hopefully the issue is clearer when seeing the Dirichlet KL mathematically, as the summation over classes is what causes this issue. On a toy example, we show the scaling issues as dimensionality grows. We weren’t sure if ‘weaker’ referred to the prior $\alpha$ values or KL annealing term. We tested both, and showed that KL annealing is the better strategy when using a linear scaling heuristic.
> 4. **Local weight priors.** We were not familiar with this paper and the results (their Fig. 4) certainly look very interesting. In the Fig. 3 caption and Section 4.1 we were referring to the corresponding weight priors (i.e. LHS of Fig. 3), not weight space priors in general. The SNGP paper we cite at the end uses weight space priors and RFF features. We’ve adjusted the sentence in Section 4.1 to avoid confusion.
> 5. **Full fKL gradients.**  This is an interesting aspect that we considered a lot when developing this idea. The approach presented in this submission is motivated by seeing if the simpler approximation worked well enough. One option we considered was using the Implicit Function theorem (mentioned in the bi-level optimization review linked). We are still interested in this idea, but we haven’t had time to implement it yet, and we feared it would add a lot more complexity to the paper. The second option is backpropagation through the Dirichlet parameters, but this was impeded by Pytorch not (yet) supporting polygamma functions of order n >= 2. Again, we are also interested in implementing this approach but have not had time to look into it. Our hope is that we could include a study of the approximation in a camera-ready submission if this submission is accepted.
>
> We hope this has addressed your concerns. Please let us know if you have any further questions or concerns.

---

> > ### Comment · Reviewer_7nPR · 2021-11-30
> > **brief response**
> >
> > Classes:
> > The effect of a weakened KL-term and a weaker prior (i.e. a prior with less pseudo-counts) are not mathematically equivalent in general, but would have similar effects in this case. Might be worthwhile trying.
> >
> > Weight priors:
> > SNGP does not really use weight priors, it combines spectral normalization with random features to approximate a GP AFAIK, so I am somewhat confused. When I refer to global priors I refer to standard N(0,sigma) priors as in most Bayesian NNs.
> >
> > Implicit processes:
> > I understand. As noted in my review, I tried to not let this impact my judgement of the paper, but I found it less intuitive than just reconstructing the story in isolation here, but I will leave it to the authors to chose their preferred narrative, of course.
> >
> > full gradient:
> > That is unfortunate. I was looking forward to this! For implementations in future work if fully commited to pytorch-land, the authors could look into libraries such as higher https://github.com/facebookresearch/higher to implement this without too much pain.
> > Higher will use a context manager to 'emulate' a functional setup (similar to jax in some sense) where gradients are carried forward. It might reduce your implementation pain by a lot.

---

### Author Response · Authors · 2021-11-19
**Rebuttal Overview**

Generally the reviewers said the work was relevant to the conference, clear, timely and interesting. The main issues concerned ablation studies to understand the role of hyperparameters, the implications of our approximations and novelty w.r.t. prior work.

We have submitted a new draft that contains several improvements thanks to the reviewer feedback (highlighted in magenta).

**Improvements**

**Clarity.** Reviewer rF5E pointed out that we did not discuss the batch aspect of the fKL estimation and our factorized prior and posterior assumption. We have discussed this additional approximation to the submission, and fixed errors in Eq. 5 and 6.
In prior work, this batch aspect is important because rich GP priors are used that capture interesting correlations between points. For our work, as our focus is larger scale image tasks and we’re working with Dirichlets, we opt for simpler factorized priors and posteriors in order to simplify computation and prior design. This factorization assumption had been shown to work in the prior work discussed in Appendix A.
We also noticed there is a discrepancy in how we discussed index and measurement sets compared to Sun et al, so we added some additional clarifying statements in this regard to improve clarity.

**Novelty.** Several reviewers questioned the novelty of the work compared to Belief Networks of Joo et al. Belief networks correspond to our MAP fVI. The novelty of this work is connecting the approach to fVI, which allows us to work with stochastic BNNs and motivates designing the index set for OOD performance.

**Baselines.** We have added a comparison to Belief Matching and Prior Networks. We used the open source implementations on the CIFAR10 corruption task. Unfortunately we could not reproduce the Prior Network results with their code and hyperparameters, and the Prior Network authors have yet to reply to our questions regarding this issue.

Toy examples and ablations on label dimension, annealing and prior design were also added to answer authors' questions.

---

### Public Comment · ~Pranav_Poduval1 · 2022-08-08
**Interesting work. Kindly check the similar prior work "Functional Space Variational Inference for Uncertainty Estimation in Computer Aided Diagnosis"**

Dear Authors,
This is an interesting work. However I urge you to have a look at a similar prior work - https://arxiv.org/abs/2005.11797

We request you to cite the same.

Thank You

---

### Decision · Program_Chairs · 2022-01-20

**Decision:**

Reject

**Comment:**

A variational function-space prior is proposed, resulting in a variational Dirichlet posterior. After rebuttal, reviewers still had many remaining questions or concerns about the paper. For instance, rF5E outlines several concerns, many relating to factorization assumptions. Reviewer 7nPR also provides several suggestions. I will not repeat them here, but do encourage the authors to look closely at these questions and suggestions. At this particular time the paper is not strongly resonating with reviewers, but could be updated so that the value of the contributions is more obvious.